# Impact of precisely-timed inhibition of gustatory cortex on taste behavior depends on single-trial ensemble dynamics

**Narendra Mukherjee[1,2,3], Joseph Wachutka[1,2,3], Donald B Katz[1,2,3]***

[1]Program in Neuroscience, Brandeis University, Waltham, United States; [2]Volen National Center for Complex Systems, Brandeis University, Waltham, United States; [3]Department of Psychology, Brandeis University, Waltham, United States

**Abstract** Sensation and action are necessarily coupled during stimulus perception – while tasting, for instance, perception happens while an animal decides to expel or swallow the substance in the mouth (the former *via* a behavior known as 'gaping'). Taste responses in the rodent gustatory cortex (GC) span this sensorimotor divide, progressing through firing-rate epochs that culminate in the emergence of action-related firing. Population analyses reveal this emergence to be a sudden, coherent and variably-timed ensemble transition that reliably precedes gaping onset by 0.2–0.3s. Here, we tested whether this transition drives gaping, by delivering 0.5s GC perturbations in tasting trials. Perturbations significantly delayed gaping, but only when they preceded the action-related transition - thus, the same perturbation impacted behavior or not, depending on the transition latency in that particular trial. Our results suggest a distributed attractor network model of taste processing, and a dynamical role for cortex in driving motor behavior.

DOI: https://doi.org/10.7554/eLife.45968.001

**\*For correspondence:**
dbkatz@brandeis.edu

**Competing interests:** The authors declare that no competing interests exist.

## Introduction

One of the primary purposes of sensory processing is to drive action, such that the source of sensory information is either acquired or avoided (in the process generating new sensory input; *Prinz, 1997*; *Wolpert and Kawato, 1998*; *Wolpert and Ghahramani, 2000*). To the extent that this is true, sensory and motor processing should be tightly coupled (*Wolpert et al., 1995*; *Huston and Jayaraman, 2011*). The gustatory system is an ideal model to study this proposed coupling, because animals necessarily respond to tastes with discriminative behaviors - specifically, they must decide to either swallow or expel the sensory stimulus in the mouth (*Grill and Norgren, 1978a*; *Katz and Sadacca, 2011*; *Li et al., 2016*).

Sensory-motor coupling is visible in the temporal response patterns of rodent gustatory cortical (GC) neurons to taste stimulus administration. GC neurons respond to taste presentation with a sequence of firing-rate 'epochs', two of which are taste-specific: neural firing first carries information regarding the physio-chemical identity of the taste stimulus, and then correlates with palatability, a variable intimately linked with the animal's decision to ingest or expel the taste (*Katz et al., 2001*; *Fontanini and Katz, 2006*; *Grossman et al., 2008*; *Piette et al., 2012*; *Sadacca et al., 2012*; *Maffei et al., 2012*; *Jezzini et al., 2013*; see also *Crouzet et al., 2015*). Ensemble analyses further reveal that the transition between these two epochs happens suddenly and coherently within neural ensembles (*Jones et al., 2007*; *Sadacca et al., 2016*). This ensemble transition to palatability coding, though highly variable in latency (between 0.5 and 1.5s post stimulus, depending on the trial), is

a strong predictor of the onset of the animal's consumption-related orofacial behavior (*Sadacca et al., 2016*), even when the timing of this behavior is manipulated by learning (*Moran and Katz, 2014*) or cueing (*Li et al., 2016*). That is, GC neural ensembles appear to 'hop' from one attractor state to another during taste processing (*Miller and Katz, 2010*; *Miller, 2016*), with the hop representing the reaching of a consumption decision - and (potentially) the emission of a motor signal to brainstem circuits that generate orofacial behavior.

A direct prediction of this temporally dynamic model of gustatory sensorimotor processing, and most specifically of the suggestion that the transition into the later firing-rate epoch represents the emission of a motor command, is that well-timed perturbations of GC activity should affect the time course of a rat's taste-reactive ingestion-egestion behavior. This prediction recently received indirect support when it was shown that optogenetic perturbation of the entire GC taste response (*Li et al., 2016*) significantly (if modestly) changes the probability of rejection behaviors in response to aversive tastes ('gapes'; *Grill and Norgren, 1978a*; *Li et al., 2016*).

However, such gross perturbations of gustatory processing are an inadequate test of this very specific prediction: for one thing, multi-second perturbations likely have secondary effects that confound interpretation, particularly regarding an outcome variable (ability to gape) that is known to depend on an interconnected network of brain regions (including GC; see *Smith and St John, 1999*; *Riley and King, 2013*; *Samuelsen and Fontanini, 2017*); in addition, it is impossible to disambiguate any epoch- or moment-specific effects on consumption behavior using whole-response perturbations. A much more definitive test would involve using optogenetics to perturb GC taste responses for short periods of time as awake rats process and respond to a range of tastes.

Here we report the results of precisely this experiment, performed in awake, tasting rats. We recorded the activity of GC ensembles while simultaneously perturbing the firing of these neurons using an optogenetic silencer (specifically, the proton-pump ArchT) for brief (0.5s) periods before, during or after the 'hop' to the palatability- (i.e., decision-) related state. Our results provide strong support for the hypothesized importance of the transition time itself, and in addition suggest that important pre-transition taste processing is performed within GC (these results reveal limitations on the validity of time-free interpretations of inactivation data). Furthermore, our data provide a glimpse into the robustness of the system underlying the neural processing of taste, demonstrating that GC is one participatory node in a larger network with attractor dynamics: the fact that GC perturbations can only delay the system settling into the decision-related 'stable' state suggests that this stable state is a function of activity spread across multiple regions; in addition, the fact that post-decision perturbations have no impact suggests that behavioral control shifts to brainstem circuits once this stable state has been reached.

## Results

### Experimental paradigm and data overview

*Figure 1A* depicts the preparation used for our experients - IOCs for taste delivery, bilateral GC opto-trodes for recording of neural ensemble activity and delivery of laser light, and EMG electrodes in the anterior digastric (jaw) muscle for assaying of consumption-related mouth movements. Four weeks prior to the surgery in which we installed these assemblies, we injected AAV carrying the optogenetic silencer ArchT (along with green fluorescent protein - GFP) into GC. The GFP allowed us to confirm (post-mortem) infection of GC neurons by immunohistochemical verification of the GFP tag (*Figure 1B*).

The rats received intra-oral deliveries of 30mM sucrose (Dil Suc), 300mM sucrose (Conc Suc), 0.1mM Quinine-HCl (Dil Qui) and 1mM Quinine-HCl (Conc Qui). Most sessions involved '0.5s perturbation' trials: on 75% of the trials in these sessions, we perturbed GC firing for 0.5s, beginning either at 0s, 0.7s or 1.4s post taste delivery ('early', 'middle' or 'late' 0.5s perturbations, *Figure 1C*). These three perturbation windows tile the period containing the temporal epochs that characterize GC taste responses (*Katz et al., 2001*; *Sadacca et al., 2012*; *Sadacca et al., 2016*). More specifically, the early and late 0.5s perturbations for the most part (see below) affect GC neurons before and after the range of likely transition times into the behaviorally-relevant state containing palatability-related firing, which typically occur just before, during, or just after the middle 0.5s perturbations (*Figure 1C*, also see *Figure 1D* for a basic schematic of coding across the first 2.0s of GC taste

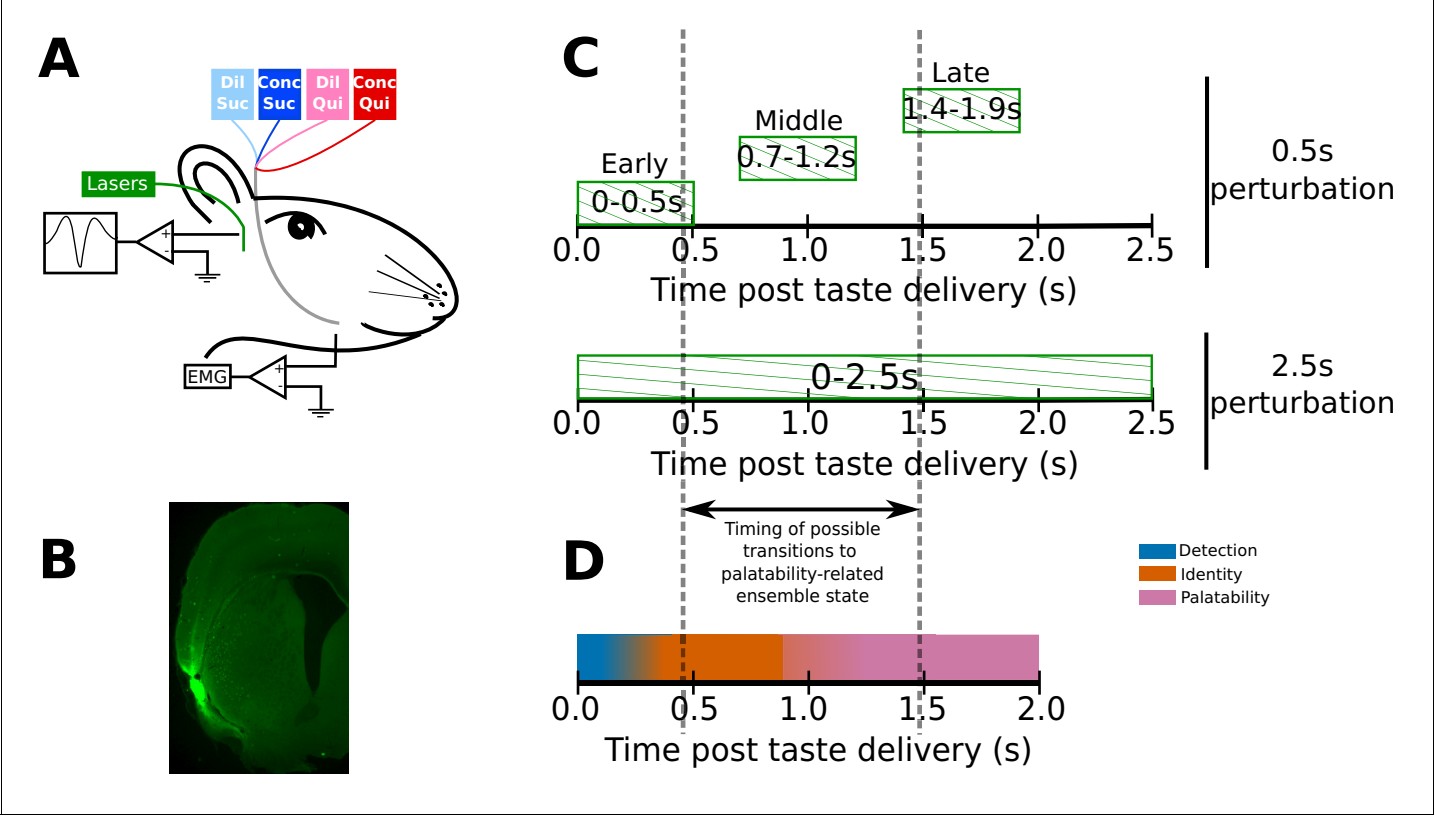

**Figure 1.** Experimental paradigm. (A) 4–6 weeks after receiving surgeries for virus injections, rats were implanted with opto-trodes and EMG electrodes. Post recovery, they were given intra-oral infusions of Dil Suc (30mM Sucrose), Conc Suc (300mM Sucrose), Dil Qui (0.1mM Quinine-HCl) and Conc Qui (1mM Quinine-HCl), and ArchT-expressing GC neurons were briefly inhibited by green (532 nm) laser light. (B) Coronal slice from a subject, showing ArchT expression (visualized by the GFP tag) localized in gustatory cortex (GC). A small lesion, left by the tip of the opto-trode is visible in the middle of the GFP expressing region, and had no general impact on behavior (see below). (C) Inhibition protocol used in the study: two types of optogenetic perturbations, short (0.5s) or long (2.5s), were delivered in separate experimental sessions. Short perturbations were delivered at one of three possible time points on any individual trial, and are referenced as follows in the text: (1) 0–0.5s - early 0.5s perturbations, (2) 0.7–1.2s - middle 0.5s perturbations, and (3) 1.4–1.9s - late 0.5s perturbations. Not shown, but delivered in all sessions, were control trials with no perturbations. Grey dashed lines mark the approximate range of the ensemble transitions to palatability/decision-related firing. (D) A schematic of the temporal structure of single-neuron coding across the first 2.0s of taste responses in GC. Immediately following taste presentation, responses are nonspecific, indicating only the presence of fluid on the tongue ('detection' epoch, in teal). The next two temporal epochs of GC firing are taste specific: the first codes the physio-chemical identity of the stimulus ('identity' epoch, in orange); following a transition (that can happen anywhere between 0.5–1.5s post stimulus on individual trials, see grey dashed lines, and on average happens midway through this period) firing rates change to reflect palatability and the upcoming consumption decision ('palatability' epoch, in pink).

DOI: https://doi.org/10.7554/eLife.45968.002

responses). A small set of additional experimental sessions (performed using a subset of the same rats) was run using 2.5s perturbations of GC (*Figure 1C*), purely as an additional comparison to the 0.5s perturbations.

We recorded the activity of 244 GC single neurons across 10 sessions (24.4 ± 13 units/session) of 0.5s perturbations, and of an additional 73 GC single neurons in five sessions (14.6 ± 4.7 units/session) of 2.5s perturbations. The two types of experimental sessions were counterbalanced, such that 3 rats received 2.5s perturbation sessions first, and 2 received 0.5s perturbation sessions first. No differences with order were noted.

The AAV-ArchT construct used in this study has been shown to infect neurons of multiple types (e.g., pyramidal neurons and interneurons) in an unbiased manner (*Aschauer et al., 2013*). Our opto-genetic protocol, therefore, is best thought of as a general perturbation of the dynamics of GC neurons in response to tastes. Any such perturbation (including those using genetics to putatively inhibit individual neuron types) will be expected to (perhaps paradoxically) enhance the firing of some

neurons through network-level effects (like disinhibition, *via* suppression of the firing of inhibitory neurons, *Allen et al., 2015*). This expectation was borne out in the data: the firing of most of our GC neural sample (146/244, 60%, example shown in *Figure 2A1–A4*) was significantly suppressed during 0.5s perturbations, but the firing of an additional 20% (49/244) was significantly enhanced. The same pattern of results was observed with 2.5s perturbations: the firing of 82% of GC neurons (60/73, example unit in *Figure 2B1–B2*) was inhibited, and the activity of 15% (11/73) was enhanced.

The fact that 2.5s perturbations appeared to inhibit a slightly larger percentage of neurons is likely an artifact caused by analysis limitations: suppression of the low firing rates (3–10Hz) that dominate GC taste responses (*Katz et al., 2001*; *Jones et al., 2007*; *Samuelsen et al., 2012*; *Kusumoto-Yoshida et al., 2015*; *Mazzucato et al., 2015*) can be difficult to detect, particularly in short time windows; consistent with this, we observed that the highest likelihood of detecting suppression during 0.5s perturbations occurred when that perturbation was delivered in the middle of taste processing (0.7–1.2s, *Figure 2C*) - at the time of peak firing rate modulations. Across 2.5s perturbations, we naturally had more statistical power to detect the suppression of firing (*Figure 2D*).

Although such perturbation protocols cannot be used to answer cell-type/microcircuit-specific questions, their network-wide effects are ideal for testing the macroscopic dynamical properties of taste processing in GC (the purpose of the current work): GC taste responses evolve through a sequence of temporal epochs (*Katz et al., 2001*; *Maffei et al., 2012*; *Jezzini et al., 2013*) which have the hallmarks of emergent, quasi-stable states of a system that can be speculatively described, at a high level, as an attractor network (*Jones et al., 2007*; *Miller and Katz, 2010*; *Mazzucato et al., 2015*; *Sadacca et al., 2016*); our protocol brings about strong perturbations of the network activity characterizing these stable states, and by mapping the state dependence of the effects of these perturbations, we are able to directly test the proposed function of these states (and of the transitions between them). Viewed in this manner, the fact that infection may have nuisance effects beyond inducing ArchT expression (and the related fact that laser illumination of GC might itself have some impact on neural firing) is of no consequence for the basic hypothesis tested below – that subtle differences in the timing of 0.5s perturbations, delivered in the same rat (and indeed, in the same session), will determine the impact of those perturbations on behavior. Thus, again, there is no need for 'empty virus' control rats.

## Early 0.5s perturbations delay single-neuron palatability-related responses while late 0.5s perturbations do not

We first assessed the impact of 0.5s (and 2.5s perturbations) on the palatability-related content of GC taste responses that had been smoothed (using 250ms-wide windows moved in 25ms steps) and standardized to be on a uniform scale (see Materials and methods for details). The set of neural responses (one per taste) were regressed against the palatability ranks of the taste stimuli (Conc Suc:4, Dil Suc:3, Dil Qui:2, Conc Qui:1) to obtain a palatability index, $\beta_{Palatability}$. Being a Bayesian analysis (consult Materials and methods for details on model setup and inference), this regression gives access to the entire posterior distribution of $\beta_{Palatability}$ at every time point. Knowing the spread of the posterior distribution of $\beta_{Palatability}$ at every time point allows us to more simply perform significance tests: we can conclude that $\beta_{Palatability}$ is different from 0 at the 1% level of significance if the 99% extent of its posterior distribution (generally known in Bayesian analyses as the 99% 'credible interval') does not overlap 0 (such time points are marked by dots in *Figure 3A1–A3, B*). We used logistic sigmoid functions to better characterize the time evolution of the posterior mean of the palatability index, $\overline{\beta}_{Palatability}$ (shown with dashed lines in *Figure 3A1–A3, B*), and defined the size and latency (time to attain 95% of maximum size) of the upper asymptote of the logistic fit as the height (*L*) and latency ($t_{peak}$) of the peak of $\overline{\beta}_{Palatability}$ respectively.

Although the impact of 0.5s perturbations on the palatability content of single-neuron GC taste responses appears modest in *Figure 3A1–A3*, this is unsurprising given the brevity of the perturbations (these perturbations also violate the assumption of data stationarity upon which curve-fitting depends, see Materials and methods for details). However, with regard to the height and timing of the peak of $\overline{\beta}_{Palatability}$ (the most robustly interpretable aspect of such an analysis), 0.5s perturbations produced dramatic and significant changes, depending on when they were delivered (*Figure 3C*).

In control (no-laser) trials during the 0.5 perturbation sessions, $\overline{\beta}_{Palatability}$ peaked ~0.8s after taste delivery. The late 0.5s perturbations (which by design started after palatability-related firing had

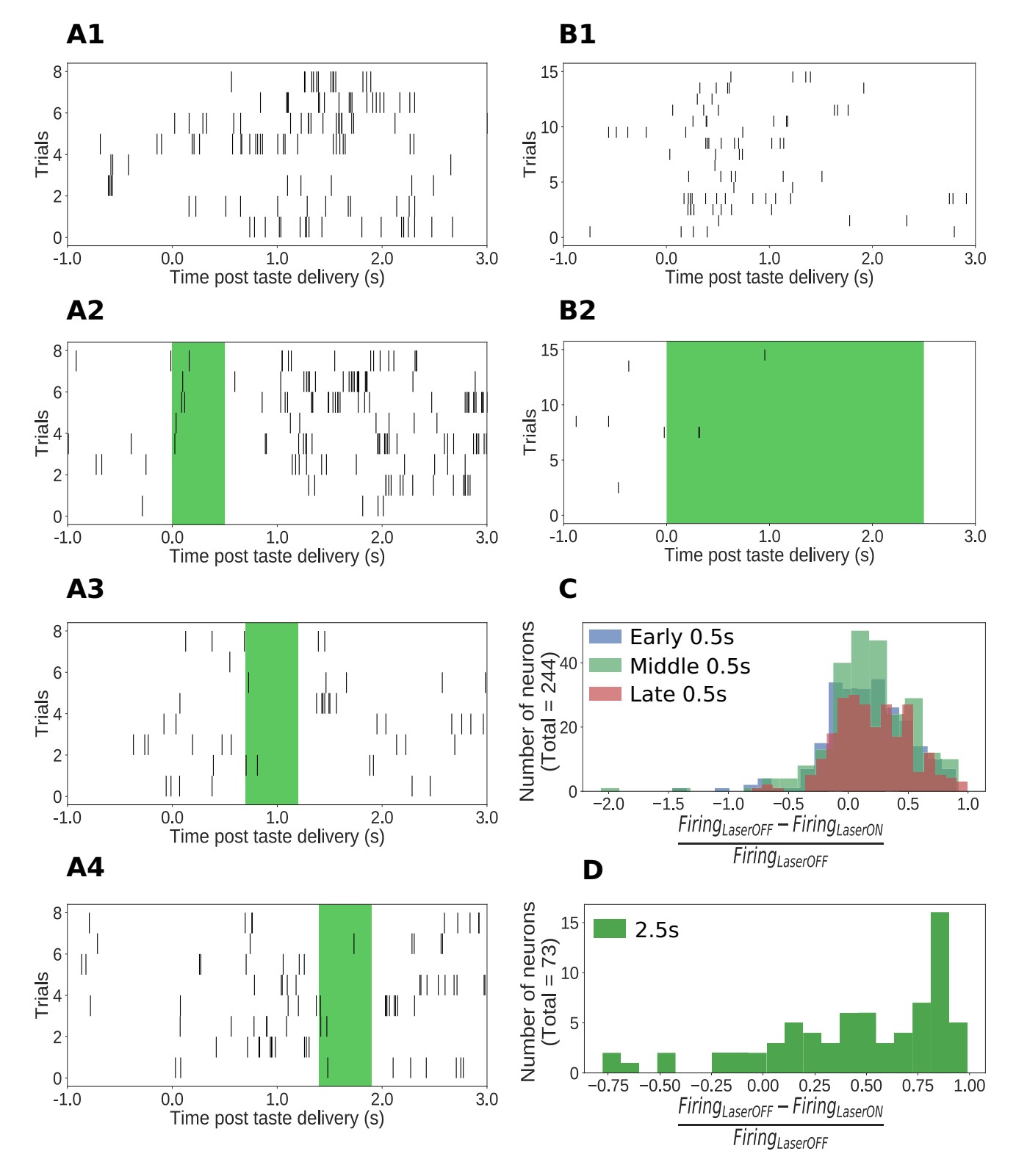

**Figure 2.** Impact of ArchT-mediated optogenetic perturbation on GC neurons. (**A1-A4**) Rasters of spiking in an example single GC neuron in a 0.5s perturbation session; each hash mark is an action potential. Activity is robustly suppressed during laser stimulation. (**B1-B2**) Analogous data from an example single GC neuron in a 2.5s perturbation session, also showing clear inhibition during laser stimulation. (**C**) Histogram of changes in firing rates (plotted as a fraction of the firing rate on control trials, x-axis) produced by 0.5s perturbations across the entire sample (y-axis = number of neurons).

*Figure 2 continued on next page*

*Figure 2 continued*

The majority of neurons show robust firing suppression when perturbed (fraction > 0), but a small group of neurons actually increased their firing rates in response to perturbation, presumably due to network-level effects (fraction < 0). (D) Analogous histogram of changes in firing rate produced by 2.5s perturbation. Almost all neurons were affected by the perturbation: the large majority are suppressed, but a small minority show elevated firing rates in response to perturbation.

DOI: https://doi.org/10.7554/eLife.45968.003

emerged on most trials) sharply reduced the palatability index only after the lasers were switched on at 1.4s (*Figure 3A3*); these perturbations did not impact the timing of peak palatability firing, although the non-stationarity of these firing data coupled with the susceptibility of standard Gaussian-noise based curve fitting techniques to outliers (see Materials and methods) resulted in a slight, artifactual shift in the function's $t_{peak}$ (happening slightly earlier than in control trials, blue bars in *Figure 3C*) which could be entirely attributed to the drop in $\overline{\beta}_{Palatability}$ after 1.4s.

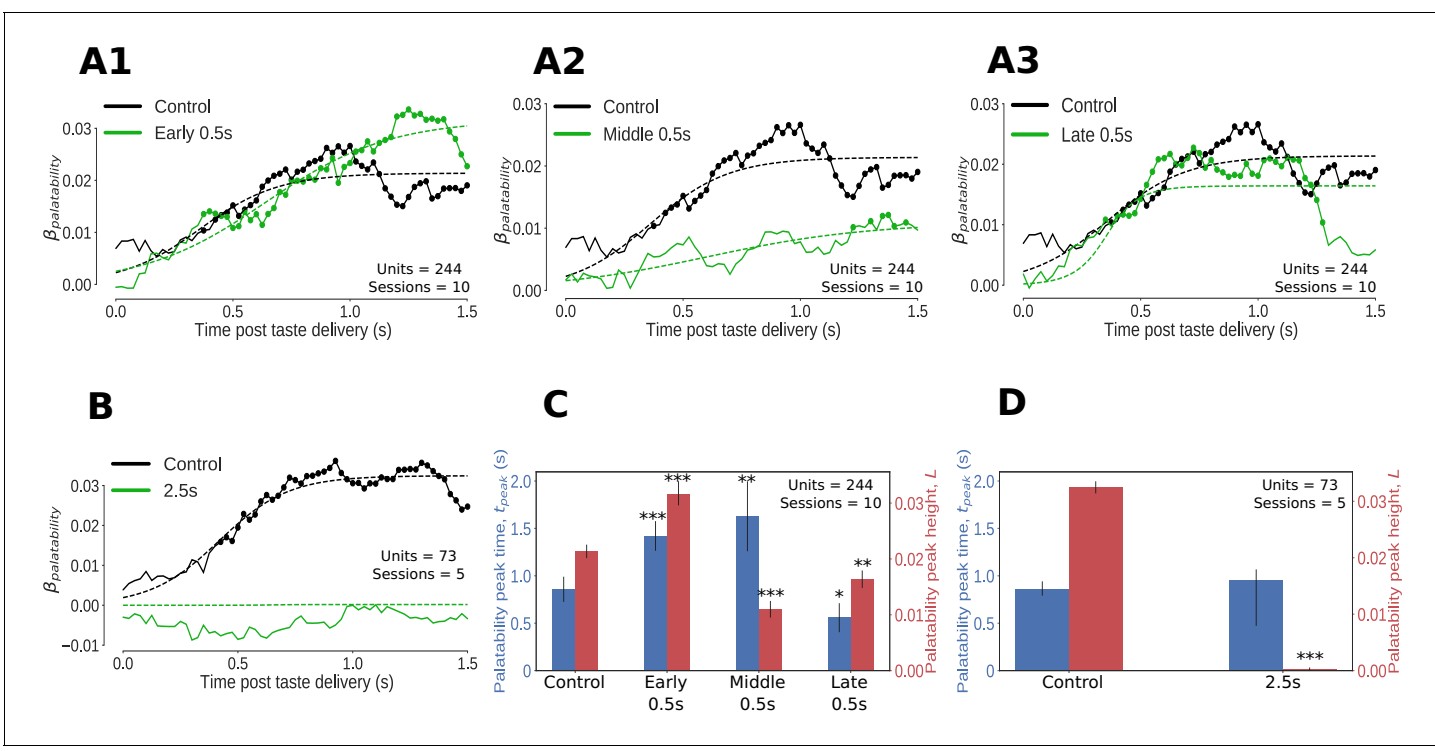

**Figure 3.** Impact of optogenetic perturbations on palatability relatedness of the firing of GC neurons. (A, B) Coefficients (palatability-relatedness, y-axis) obtained from the regression of trial-averaged firing rates on palatability ranks of the taste stimuli across time (x-axis). The solid lines depict the mean regression coefficient across time for the entire data sample; coefficients significantly different from 0 at the 1% level are marked by dots. The dashed lines are logistic sigmoid fits for each condition. A1-A3 display the impact of 0.5s perturbations beginning at 0 (early), 0.7 (middle) or 1.4s (late) on palatability-related firing in GC; B shows the analogous graph for 2.5s perturbations of GC neurons - disruption of GC firing for 2.5s wipes out the entirety of the palatability response. (C, D) The post-stimulus latency (blue bars and y-axis) and height (red bars and y-axis) of the peak (95% of the asymptote) of the sigmoid fits in A, B. Error bars denote 95% Bayesian credible intervals; each optogenetic condition was compared with control (laser-off) trials from the same sessions (first pair of bars in both figures) and statistically significant differences are marked with asterisks (*: p<0.05, **: p<0.01, ***: p<0.001). (C) On control (no-laser) trials, GC neurons asymptote to peak palatability firing ~0.8s post stimulus. Early and middle 0.5s perturbations delay the peak of palatability firing by ~0.5s; the magnitude of this peak, however, is the smallest for the middle perturbation. Late 0.5s perturbations, beginning after palatability-related firing has mostly subsided, produce a small change in the magnitude and timing of its peak compared to control trials - though significant, this minor shift in peak palatability firing can be entirely attributed to the decrement in the palatability index only after the laser is switched on at 1.4s (A3). (D) During 2.5s perturbation sessions, GC neurons asymptote to peak palatability firing ~0.8s post stimulus on control trials where the lasers are off, just like the 0.5s control. The 2.5s perturbation, by disrupting the palatability response completely, is fit by a flat sigmoid whose peak magnitude overlaps 0, although the latency to peak is similar to that of control (no-laser) trials.

DOI: https://doi.org/10.7554/eLife.45968.004

In contrast, early 0.5s perturbations caused the palatability content of GC taste responses to be delayed in reaching its peak; the function did not asymptote until ~1.3s, a lag of almost 0.5s compared to control (no laser) trials (*Figure 3A1*, and blue bars in *Figure 3C*). Note that the failure of GC firing to 'bounce back' immediately after laser-off (which occurred 300–400ms before the time of peak palatability content in control trials) implicates GC in the processing of palatability itself (see Discussion). Note as well that despite delaying the peak of $\overline{\beta}_{Palatability}$, the magnitude of this eventual peak was actually higher than control trials (red bars in *Figure 3C*). The early 0.5s perturbations thus appear to cause a transient shift out of the attractor dynamics responsible for GC taste responses followed by gradual relaxation back into the stable state after the end of the perturbation; variability in this process (which can overshoot the stable point, depending on the speed of relaxation) could explain the apparent increase in the magnitude of the peak palatability index in this condition.

Middle 0.5s perturbations (*Figure 3A2*) also had a powerful impact on GC palatability-related firing: compared to no-laser trials from the same sessions, the peak of $\overline{\beta}_{Palatability}$ emerged later (blue bars in *Figure 3C*) while its magnitude was largely reduced (red bars in *Figure 3C*). The potency of both these effects makes sense, as this particular perturbation overlaps the heart of palatability-related activity in GC neurons (*Katz et al., 2001*; *Jezzini et al., 2013*; *Sadacca et al., 2016*).

Finally, as expected, 2.5s perturbation had a devastating impact on palatability-related responses of neurons in the affected GC network (*Figure 3B*). In control (no-laser) trials, as in previous studies (*Sadacca et al., 2016*) and no-laser trials during the 0.5s perturbation sessions, $\overline{\beta}_{Palatability}$ climbed to an asymptote ~0.8s after taste delivery. Meanwhile, 2.5s perturbation effectively inhibited any significant rise in $\overline{\beta}_{Palatability}$ (red bars in *Figure 3D*), which is unsurprising given the drastic impact of these perturbations on neural firing in GC (*Figure 2*).

## GC perturbation delays the onset of aversive orofacial behavior

We monitored our rats' mouth movements *via* electromyography (EMG). Specifically, we implanted EMG electrodes in the anterior digastric muscle; as a jaw moving muscle, the anterior digastric plays a major role in the production of 'gapes', the rhythmic orofacial behavior that serves to move aversive tastants to the front of the mouth in preparation for expelling. Far less accessible tongue muscles underlie mouth movements that support behaviors (such as 'lateral tongue protrusions') that help the rat prepare to ingest appetitive tastants (*Grill and Norgren, 1978a*; *Travers and Norgren, 1986*; *Li et al., 2016*). For that reason, we focus solely on gapes in this study (but see Discussion).

Individual mouth movements can be recognized as bursts of anterior digastric EMG activity (*Figure 4A1–A2*). However, the variability in the amplitudes and durations of these EMG bursts reduces our ability to separate gapes from other large mouth movements. We therefore made use of a more robustly distinctive feature of gaping – the fact that gapes occur in 4–6Hz bouts (*Travers and Norgren, 1986*; *Li et al., 2016*). We analyzed the spectral content of the envelope of the EMG signal using Bayesian spectrum analysis (BSA; see Materials and methods for a detailed discussion) and measured the probability of gaping as the total posterior probability of 4–6Hz movements.

While easier to calculate and less subject to error, this estimate of the probability of gaping has strong correspondence with gaping bouts identified by a classifier trained on individual bursts of EMG activity (*Li et al., 2016*), note in *Figure 4B1–B2* how the calculated probability of being in a gaping bout corresponds to the classifier-estimated timing of individual gapes, shown as vertical hash marks); the trial-averaged probability of gaping calculated by BSA is similar to the peri-stimulus 'rate of gaping' obtained from the gapes identified by the classifier, for both trial types in which gaping occurred (Dil and Conc Qui trials, *Figure 4C1–C2*). Finally, the fact that the probability of gaping jumps precipitously just before the first gape as identified on video (*Figure 4D*) confirms this algorithm's reliability in identifying periods of gaping in the EMG signal (see Materials and methods for more details).

With this information in hand, we were able to investigate the effects that perturbations of GC activity have on the animals' rejection of aversive Qui. On average, gaping begins ~0.9s after Qui delivery in control trials during the 0.5s perturbation sessions – that is, when analysis is restricted to trials in which GC neurons were not perturbed (*Figure 5A*, inset). This latency is consonant with that reported in video analysis (*Grill and Norgren, 1978a*) and classic burst-oriented analysis of EMG

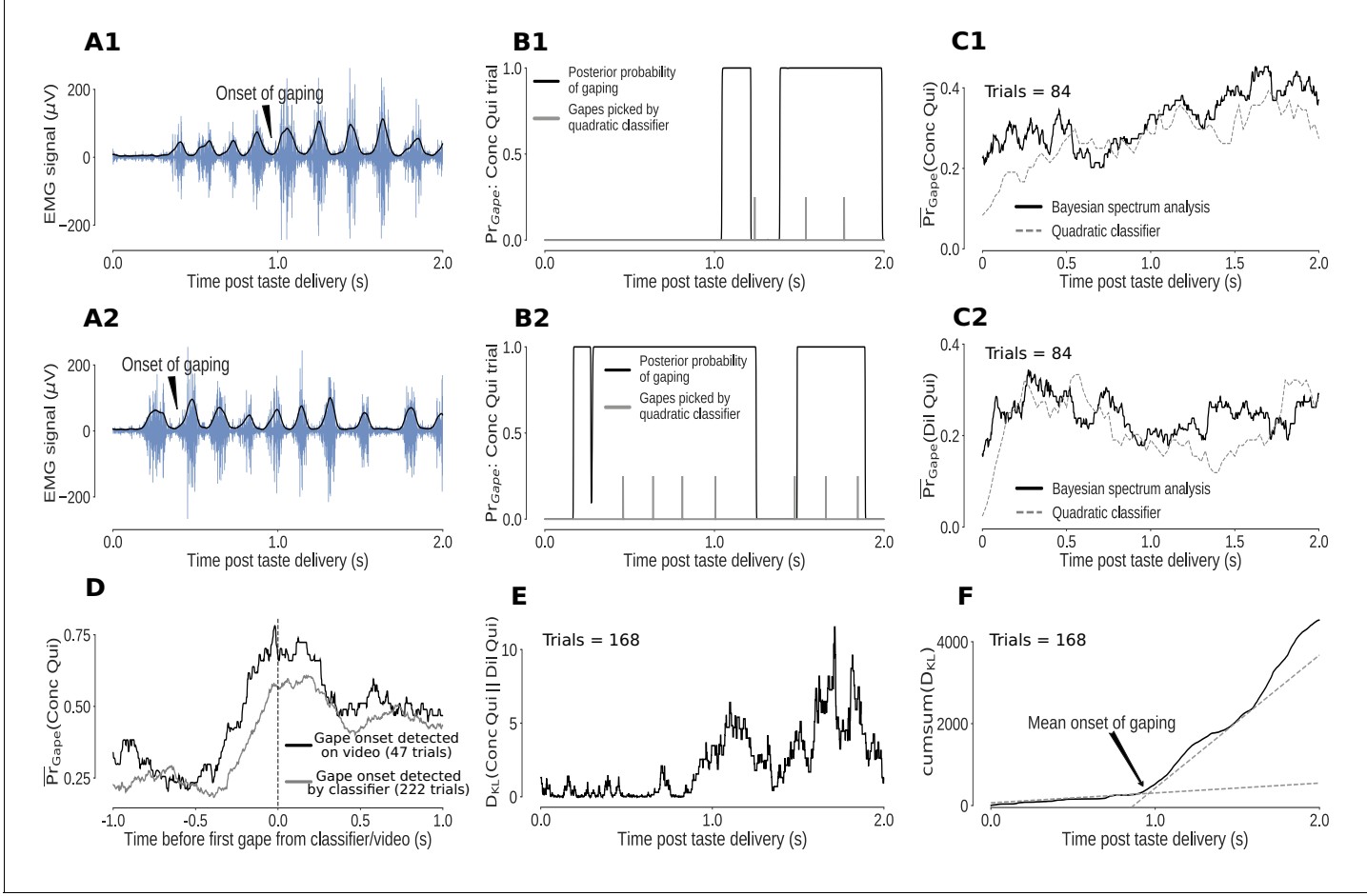

**Figure 4.** Bayesian spectrum analysis (BSA) of anterior digastric EMG recordings - probability of gaping calculated in terms of the total posterior probability of 4–6Hz movements. (**A1-A2**) Two representative Conc Qui trials. The animal's mouth movements can be seen as bursts of higher-amplitude (y-axis) EMG activity (blue) following taste delivery - the onset of gaping, as detected on video, is marked. The time series of the envelope of the EMG signal (black line) are the data subjected to BSA. (**B1-B2**) Result of BSA brought to bear on a pair of individual Conc Qui trials. The calculated probability of gaping (y-axis, black lines) matches up with individual gapes (grey vertical hash marks) picked by a previously published quadratic classifier that achieved 75% accuracy. (**C1-C2**) The trial-averaged probability of gaping (across a set of no-laser control trials) calculated with BSA (solid line) matches up with the peri-stimulus 'gaping rate' produced from the gapes identified by the quadratic classifier (dotted line, same set of control trials) in response to both Dil Qui (**C1**) and Conc Qui (**C2**). While correlating well with the quadratic classifier, BSA avoids multiple pitfalls of that technique (and is easier to apply, see Materials and methods). (**D**) The probability of gaping calculated with BSA rises reliably just before the first gape, detected either on video (black) or by the quadratic classifier (grey). The black dashed line (0 on the x axis) indicates the time of the first gape. (**E**) KL divergence between the probability of gaping to Conc and Dil Qui (higher values indicate larger differences in their gaping distributions, same trials as in (**C1, C2**). As expected, the distributions of gaping probability on Conc and Dil Qui trials are initially similar (while non-specific investigative licks happen) and diverge out at ~1s post stimulus once gaping begins. (**F**) The cumulative sum of the KL divergence in E across time. The jump in KL divergence around the mean onset time of gaping is seen as a change in slope of its cumulative sum. We fit two straight lines to the cumulative sum and pick their intersection as the mean onset of gaping across this set of trials.

DOI: https://doi.org/10.7554/eLife.45968.005

(*Travers and Norgren, 1986*). Furthermore, this estimate matches up with observations in control rats (published in *Sadacca et al., 2016* and *Li et al., 2016*) that received neither laser nor ArchT expression. Thus we can conclude that, at least with regard to the driving of gaping, our preparation leaves the system capable of normal function – neither infection nor implantation themselves affect the ability to gape to Qui.

Previous work has shown that while the appearance of palatability-related firing in GC (which arises suddenly and coherently across neurons in single trials) robustly predicts the onset of gaping bouts (see below and *Sadacca et al., 2016*), it is unrelated to the mechanics of individual gapes

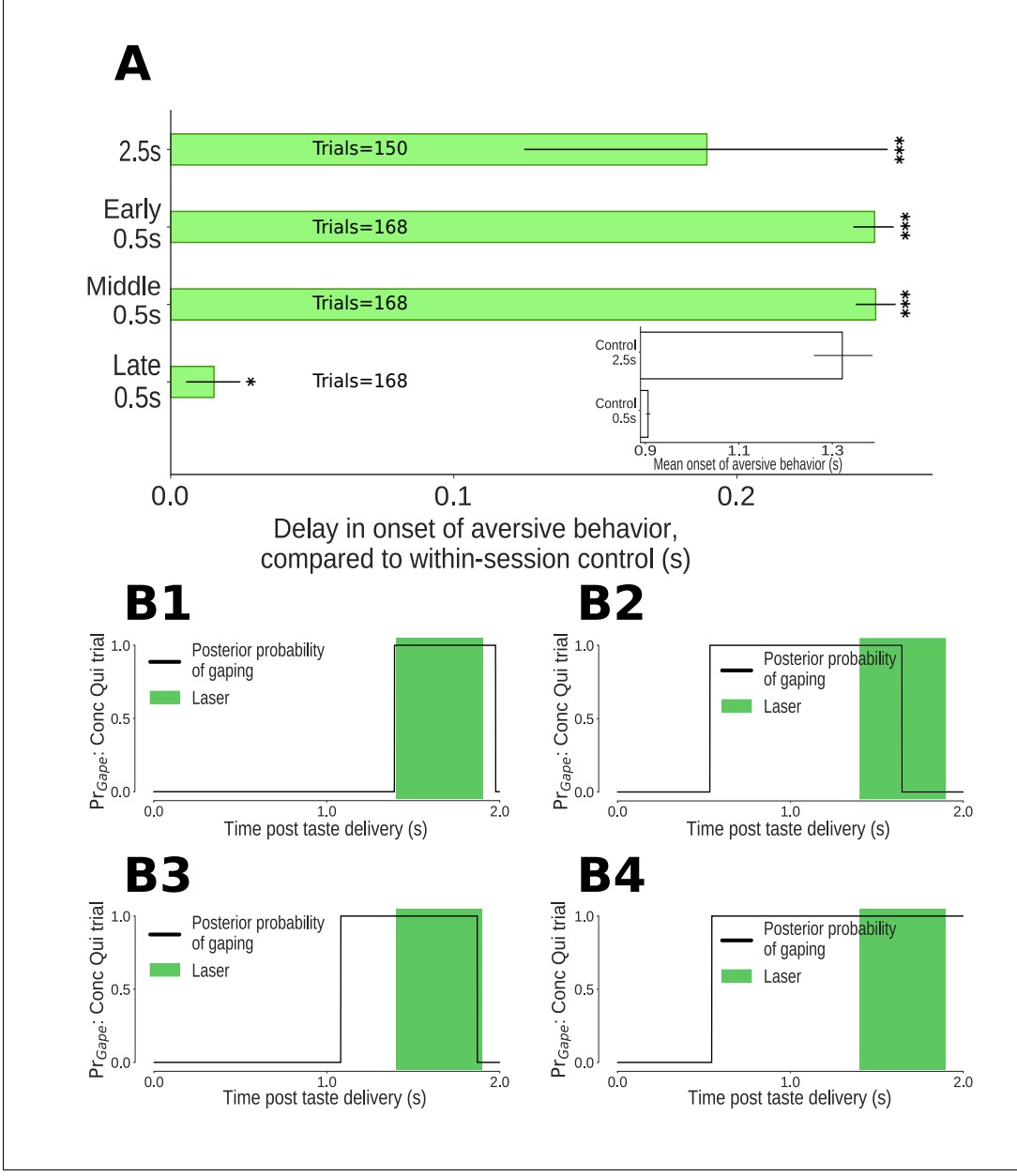

**Figure 5.** The impact of 0.5s perturbations of GC neurons on gaping (4–6Hz aversive orofacial behaviors) depends on when they were delivered. (**A**) Onset times of gaping under different perturbation conditions, reported as delay in onset times compared to control (no-laser) trials from the same sessions. The x-axis presents the mean delay in gape onset times compared to control trials; the extent of their 95% Bayesian credible intervals are shown by the error bars. We indicate statistically significant delays with asterisks (*: p<0.05, **: p<0.01, ***: p<0.001). Early and middle 0.5s perturbations of the taste response delay the onset of gaping (to the same degree as 2.5s perturbation). Late 0.5s perturbations of GC neurons produce a minor (but marginally significant) delay in the onset of gaping. (**A, Inset**) Latency of gaping in control (no-laser) trials in 0.5s and 2.5s perturbation sessions. The 2.5s controls show a delayed onset, likely due to lasting effects of the (relatively) long optogenetic perturbation. (**B1-B4**) Four representative Conc Qui trials with late 0.5s perturbations (1.4–1.9s post taste delivery). On each of these trials, the probability of 4–6Hz aversive orofacial responses is unaffected by the onset of the laser, confirming that GC perturbation fails to disrupt ongoing bouts of gaping.

DOI: https://doi.org/10.7554/eLife.45968.006

within gaping bouts (*Grill and Norgren, 1978b*; *Li et al., 2016*). We therefore predicted that late

0.5s perturbations – perturbations delivered once gaping was (in most trials, see below) already underway – would have minimal impact on gaping behavior.

Our data are in accord with this prediction. Rats gaped with near normal latency, compared to control (no laser) trials, in late 0.5s perturbation trials (*Figure 5A*, but also see below for discussion of the marginally significant delay in gape onset in this condition). Furthermore, these late 0.5s perturbations failed to prematurely end gaping bouts that had already begun. *Figure 5B1–B4* show example trials in which the probability of gaping rhythm in the EMG signal went high following Conc Qui delivery, and stayed high throughout late 0.5s perturbations. In fact, the percentage of trials in which gaping was maintained into this period was unchanged by late 0.5s perturbation - 57% (36/63) of control trials vs 55% (26/47) of perturbed trials. We can thus conclude that GC is of no consequence for the maintenance of ongoing gaping.

In contrast, GC activity plays a clear role in the initiation of gaping: early 0.5s perturbations – that is, those delivered well before transitions into the palatability-related state of GC activity – delayed gaping onset by approximately 0.25s on average (*Figure 5A*). This delay cannot be explained in terms of inhibition of the earliest gapes - gaping latencies as early as 0.5s after taste delivery were rare, and an analysis of control (no-laser) trials from the same sessions showed that removing latencies of less than 0.5s had essentially no impact on the mean onset time of gaping. The much more likely explanation is that early 0.5s perturbation of GC perturbs the ongoing process that leads to the release of a 'decision to gape' signal visible in GC (*Sadacca et al., 2016*).

Similarly, middle 0.5s perturbations of GC delayed the onset of gaping until just before 1.2s after taste administration - approximately 0.25s after gaping on no-laser trials. That is, 0.5s perturbations of GC activity occurring either before or during the 'heart' of quinine processing had a strong impact on the latency of aversive orofacial behavior. Not only is the impact of this brief optogenetic perturbation significant, it was every bit as large as that observed with whole-trial (i.e., 2.5s) perturbations, which delayed the appearance of gaping by ~0.2s (*Figure 5A*). These long perturbations are not discussed further, because: 1) they are of little relevance to our primary hypotheses; and 2) they had the additional unintended consequence of impacting gaping behavior on control trials (see *Figure 5A* inset and Discussion).

## GC perturbation impacts orofacial behavior only if delivered before the coherent ensemble transition to palatability-related activity

Previous demonstrations that the temporal dynamics of GC taste responses are well described in terms of sudden transitions between stimulus-specific ensemble firing rate 'states' (*Jones et al., 2007*; *Sadacca et al., 2016*) also reveal large trial-to-trial variability of both the behavioral and neural transition latencies. These studies show that the neural transition can potentially happen at any of a range of latencies spanning the approximate interval between 0.5 to 1.5s, and that orofacial behavior follows close behind, such that trial averaging smears the changes in firing rates into a more gradual-seeming ramp. Our middle 0.5s perturbations are delivered in the midst of the period in which transitions into this palatability-related ensemble activity state are most likely, but variability in the timing of the ensemble state ensures that there will inevitably be a subset of middle 0.5 perturbation trials in which the ensemble state transition occurs before the perturbation.

This fact affords us an opportunity to make and test a strong prediction: we predicted that identical middle 0.5s perturbations would impact gaping latency differently depending on whether, in that specific trial, the neural transition into the palatability-related ensemble activity state had already occurred – that a middle 0.5s perturbation will have no impact on a trial in which the transition has already occurred, and will have a strong impact on a trial in which it has not. This prediction implies that the results in *Figure 5A*, averaged across all trials receiving the middle 0.5s perturbation, mask the true diversity of that perturbation's possible effects.

We have previously used Hidden Markov Models (HMMs) to detect ensemble firing rate transitions in GC responses to tastes (*Jones et al., 2007*; *Moran and Katz, 2014*; *Sadacca et al., 2016*), but this analysis is not amenable to the data in this current study: a dataset made up of all four trial types (early, middle, and late 0.5s perturbations, plus no-laser control trials) would be complex enough (each trial type would likely involve distinct sets of firing rates, see below) that the HMM would be unlikely to reach stable solutions; divided into individual trial types, meanwhile, the datasets would be too small to allow convergence to even simple stable fits. Instead, we took advantage of the insights gained from our previous publications (*Katz et al., 2001*; *Fontanini and Katz, 2006*;

*Jones et al., 2007*; *Grossman et al., 2008*) and built a constrained change-point model of GC population activity; specifically, the model consisted of 2 activity change points, the latter of which introduced palatability-related firing. This model constrained the general HMM framework in a way that allowed us to estimate transitions in individual trial types (see Materials and methods for details).

*Figure 6* schematizes the change-point analysis as brought to bear on GC ensemble taste responses. The top row of the Figure shows control (no-laser) trials, in which (as described earlier) GC taste responses progress through a sequence of firing-rate states, the latter two of which are taste-specific – the former of these two reflecting taste identity and the latter reflecting taste palatability (and thereby predicting behavior); the fact that the change points ($C_I$ and $C_P$) between these states can happen at a range of latencies in different trials is reflected in the fact that the color representing one state gradually changes to the color representing the next. The next two rows represent early and late 0.5s perturbation trials, respectively: early 0.5s perturbation trials are schematized as pushing the change points back in time, although it is beyond the scope of this manuscript to delve into the complex and difficult issue of precisely how 0.5s perturbation changes the fine structure of post-perturbation activity; meanwhile, both $C_I$ and $C_P$ typically precede late 0.5s perturbations (but see below).

Finally, the last two rows represent the two possible situations involving middle 0.5s perturbations – one in which $C_P$ occurs prior to the perturbation, and one in which it does not. Our prediction is that, despite the fact that the middle 0.5s perturbation is identical in the two trials, the impact of that perturbation will be sharply specific to trial-type – gaping will be delayed in the latter, but not in the former.

We used the change-point model to identify the putative transition times into the palatability-related ensemble state for control (no-laser, grey) Qui trials, and for middle 0.5s perturbations trials (green, *Figure 7A*). Note that we could not, of course, identify change-points when the lasers (which by design altered firing) were on, and that the concatenation of pre- and post-laser responses for purposes of the analysis led inevitably (as expected, also see Materials and methods), to a higher-than-chance 'pile-up' of change points being artifactually identified at the 'seam' joining the edges of the excised time period; for this reason, with exceptions noted below, we ignore the uninterpretable change points identified in the two bins (50ms) preceding laser onset for subsequent analyses.

Transitions into palatability-related activity ($C_P$) occurred prior to perturbation in 55% of the analyzed trials (92/168). Regression analysis revealed that, on average, significant palatability-related information appeared in single neuron firing during trials in which the ensemble state transition occurred prior to the perturbation; this information was notably lacking in trials in which the transition had not occurred by perturbation onset (*Figure 7B*).

Next, we applied the outcome of this analysis to the behavioral data (*Figure 7C*): overall, middle 0.5s perturbations delayed gaping by ~0.25s (data reprinted from *Figure 5A*); on trials in which that perturbation arrived before the ensemble transition, however, the delay in gaping was far larger (more than ~0.5s) - on these trials, gaping onset was delayed until more than 0.2s after the end of the perturbations. A comparison with control trials (as always, no-laser trials from the same sessions) confirmed that this effect was not caused by a simple truncation of the distribution of gaping latencies: even when we restricted ourselves to analyzing only the proportion (31%, 52/168) of control trials which lacked any gaping-related EMG activity until 1.2s (which was, in middle 0.5s perturbation trials, the laser off time), the average gaping latency was still significantly less then that observed in the (larger) subset (45%) of middle 0.5s perturbation trials in which the ensemble transition failed to precede perturbation onset. Clearly, 0.5s perturbations delay gaping to Qui if that perturbation begins prior to the ensemble neural transition into palatability coding.

When we restricted our analysis to middle 0.5 perturbation trials in which $C_P$ happened at 0.65s or earlier (that is, prior to perturbation onset, see above), meanwhile, the onset of gaping was not delayed. In fact, gaping on these trials occurred more than 300ms earlier than in control trials – the predictable result for trials in which the neural process of reaching a decision to gape completed early. These data demonstrate that, once this decision (which clearly involves this GC activity, see above and *Sadacca et al., 2016*) has been made, 0.5s perturbations have no impact on the execution of the decision. As predicted, middle 0.5s perturbations could have either a strong effect or no effect, depending on whether the GC ensemble transition into palatability-related firing had already occurred.

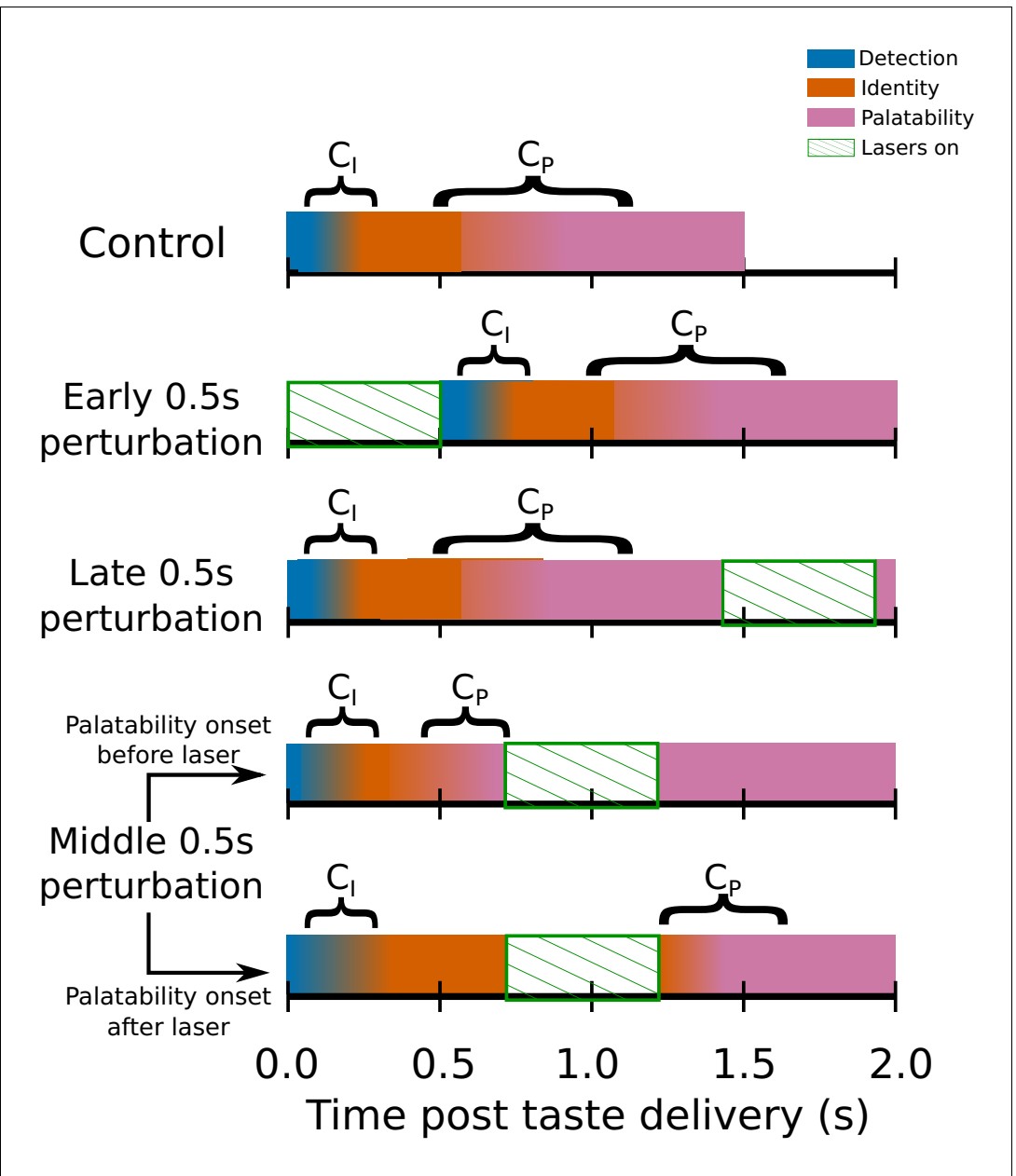

**Figure 6.** Switchpoint model of GC ensemble responses to tastes, which were assumed (on the basis of our previous work) to consist of 3 states of population firing as follows: (1) **Detection:** a brief, initial state of nonspecific responses with identical population distributions of activity for each tastant in our battery; (2) **Identity:** responses related to the chemical identity of the taste stimulus with two population firing rate distributions, one each for Suc and Qui; (3) **Palatability:** population firing rich in palatability and consumption-decision related information with four population distributions of activity, one for each of the four tastants in our stimulus battery. $C_I$, the first change point in the model denotes the shift from detection to identity firing, while $C_P$ marks the transition between the identity and palatability-related ensemble states. The fact that both $C_I$ and $C_P$ have massive trial-to-trial variability is denoted by the colors for each of the three states gradually blending across time. The model assumed that $C_I$ and $C_P$ could not occur during the optogenetic perturbation of GC (denoted by periods of green diagonally hatched regions): each row shows how the search for change points is hypothesized to be impacted by GC perturbation; note the two distinct possibilities with regard to middle 0.5s perturbation (lasers on from 0.7s to 1.2s post stimulus) trials.

DOI: https://doi.org/10.7554/eLife.45968.007

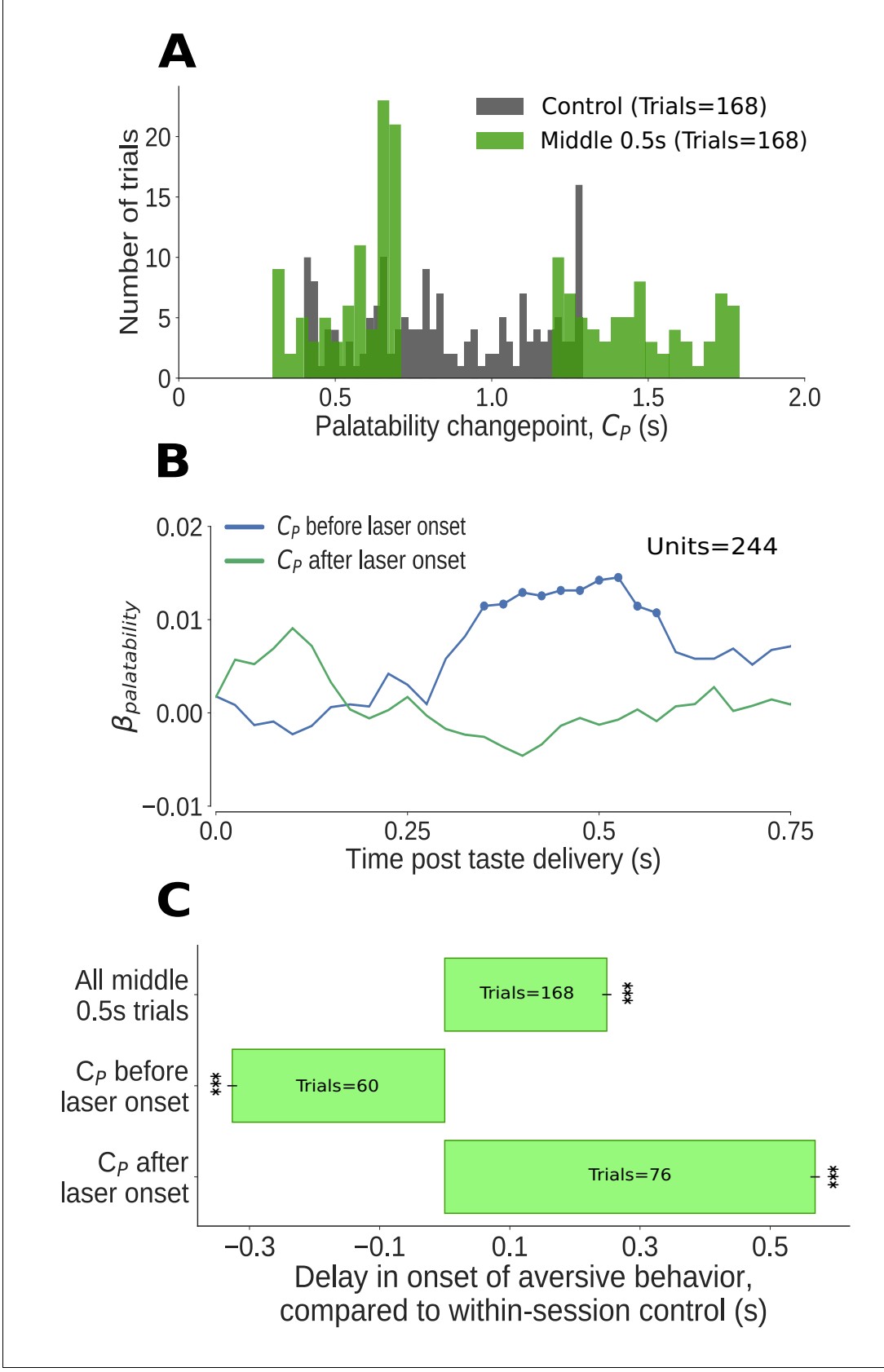

**Figure 7.** The impact of middle 0.5s perturbation (0.7–1.2s post taste delivery) on GC neural activity and aversive
*Figure 7 continued on next page*

*Figure 7 continued*

orofacial behavior varies from trial to trial, depending on the progress of taste dynamics. (**A**) Distribution of change-points into the palatability-related ensemble state ($C_P$) identified in Qui trials (in green). We could not examine the time period of perturbation, from 0.7s to 1.2s, because firing during this period was deeply confounded by laser-induced inhibition: as we concatenated the pre- and post-perturbation periods, an abnormally large number of change points (mostly artifacts, see text for details) are localized to the time of splicing (compare to change points identified in control trials, in grey). Due to these artifacts, we ignored trials with change points within 50ms of laser onset (i.e, between 0.65s and 0.7s). (**B**) Correlation (quantified in terms of coefficient of regression) of trial-averaged firing rates of GC neurons with palatability of the taste stimuli in two subsets of trials - those in which the ensemble transition into palatability-related firing ($C_P$) was identified to have occurred prior to perturbation (blue line), and those in which it did not (green line). Coefficients significantly different from 0 at the 5% level are marked by dots; these coefficients differ from 0 only within the trials in which the palatability-related ensemble state appeared before the onset of perturbation. (**C**) The impact of middle 0.5s perturbation on the onset of aversive orofacial behavior, quantified in terms of the delay of behavior onset compared to control trials within the same session (x-axis) (*: p<0.05, **: p<0.01, ***: p<0.001). The top bar, which is all middle 0.5s perturbations aggregated, is repeated from *Figure 5A*. The middle 0.5s perturbation delays the onset of gaping, but only if it begins before the palatability-related ensemble transition (measured by the palatability change point, $C_P$) - in fact, on trials where palatability information appears before the laser has been switched on (i.e, before 0.65s), gaping begins earlier than on control trials (as expected, see text).

DOI: https://doi.org/10.7554/eLife.45968.008

Further evidence supporting this conclusion came from examination of the few middle 0.5s perturbation trials in which the change-point analysis suggested the presence of transitions occurring between 0.65 and 0.7s. While some of these putative change points were almost certainly artifactual (see above), we found that 'early' gaping occurred in only a subset (15) of these trials - almost precisely the same number (14) as there were control trials in which the transition occurred in the 0.65–0.7s interval; this result suggests that the true transitions that occurred during this interval resulted in gaping that was unaffected by the laser perturbation.

Finally, we performed one last convergent test of our central hypothesis that 0.5s perturbations delay gaping only if delivered prior to the neural transition into palatability-related firing – a test that involved examination of late 0.5s perturbations. As can be seen in *Figure 5A*, this perturbation delayed gaping only very slightly – but significantly – compared to control (no-laser) trials from the same session. We hypothesized that this extremely modest effect might reflect the fact that, in a small subset of these trials, the late 0.5s perturbation may have preceded $C_P$, the GC ensemble transition into palatability-related firing. Change-point analysis revealed this to be the case: in a few (20/168, 12%) of the late 0.5s perturbation trials, the perturbation arrived before $C_P$. When these trials were removed from the analysis, 95% of the difference between the gaping latency in late 0.5s perturbation trials and control trials vanished.

As a whole, our results demonstrate that the impact of brief optogenetic perturbation of GC activity depends both on when that perturbation occurs and on precisely what state the brain has achieved prior to that perturbation. This result provides support for our overarching hypothesis that the onset of palatability-related population activity in GC marks a discrete shift in taste processing - the ensemble transition in taste-related firing that predicts behavior is in fact the emission of the decision to gape.

## Discussion

Sensation and action are inextricably linked in cortical taste responses. Neurons in gustatory cortex (GC), the primary sensory cortical area for taste, exhibit responses that, across 1.5s of post-stimulus time, shift from first reflecting stimulus identity to predicting a rat's consumption decision (*Katz et al., 2001*; *Fontanini and Katz, 2006*; *Sadacca et al., 2012*; *Maier and Katz, 2013*). With ensemble analysis, these otherwise gradual-seeming changes in firing rates are revealed to be swift, coherent transitions between population activity 'states' (*Jones et al., 2007*) - transitions that vary widely in latency from trial to trial, and that are therefore effectively blurred out in stimulus-aligned averages. Despite (in fact, because of) their highly variable latencies, these ensemble firing states reliably precede the onset of ingestion-egestion mouth movements by ~0.2–0.3s (*Sadacca et al.,*

2016; *Li et al., 2016*), predicting not only the nature but the latency of these movements in single trials.

Here we show that GC neural ensemble dynamics described above are not merely 'efferent copy' reflections of processes occurring elsewhere, but are instead an indication of processing that is (to at least some extent, see below) intrinsic to GC. Brief (0.5s) optogenetic perturbations of GC neurons impact the timing of the animal's decision to expel a bitter taste in the mouth, but only if those perturbations begin before the neural ensemble has shifted to palatability-related firing. Thus, a unique moment in time (the shift of population activity to reflect stimulus palatability), despite being enormously variable in latency from trial-to-trial, reflects a tipping point in taste processing; cortical disruptions have no impact beyond this tipping point, as the control of the ongoing movements themselves shifts elsewhere (presumably to brainstem pattern generators that control the ingestion-egestion mouth movements themselves in real time, see *Travers et al., 1997*; *Travers et al., 2000*).

A massively interconnected network of forebrain regions underlies or reflects taste processing - in addition to GC, this network includes the central and basolateral nuclei of the amygdala (CeA and BLA, *Nishijo et al., 1998*; *Grossman et al., 2008*; *Fontanini et al., 2009*; *Sadacca et al., 2012*), hippocampus (*Ho et al., 2011*), lateral hypothalamus (LH, *Yamamoto et al., 1989*; *Li et al., 2013*), the bed nucleus of the stria terminalis (BNST, *Norgren, 1976*; *Li and Cho, 2006*), the parabrachial nuclei of the pons (*Baez-Santiago et al., 2016*), and the nucleus of the solitary tract (NTS, *Di Lorenzo and Lemon, 2000*). Several of these brain regions have been shown to integrate sensory and motor aspects of taste stimuli in their responses (*Sadacca et al., 2016*; *Baez-Santiago et al., 2016*; *Denman et al., 2019*). Furthermore, multiple forebrain regions send direct descending feedback to the primary brainstem taste regions, influencing both their activity (*Di Lorenzo, 2000*; *Cho et al., 2003*; *Li et al., 2005*) and generation of orofacial movements (*Zhang and Sasamoto, 1990*; *Berridge and Valenstein, 1991*; *Shammah-Lagnado et al., 1992*; *Travers et al., 1997*). Given this widely distributed network of processing nodes, it is to be expected that perturbation (or disruption over long periods of time) of one (or a few) of the participatory nodes will initiate homeostatic mechanisms that minimize the resultant degradation of behavior; thus, it is unsurprising that rodents remain able to produce gapes following ablation (*King et al., 2015*) or extended disruption of GC (*Li et al., 2016*) - in fact, the basic gaping response to quinine is produced even in decerebrate rats (*Grill and Norgren, 1978b*). Nonetheless, we find that brief perturbations of GC do significantly alter these behaviors (as do lesions of other areas, such as gustatory thalamus; *Grill and Norgren, 1978b*), proving that far more than the minimal circuit is involved in triggering them in situ.

Lengthy disruptions of GC activity appear to have lasting effects that can confound the interpretation of their behavioral impact - our 2.5s long optogenetic perturbations delayed the onset of gaping even in control (no-laser) trials. Such spillover effects may reflect cellular or network-level processes, but they cannot be attributed to cell death caused by the perturbation: in our case, similar optogenetic protocols have been shown to have no observable impact on cell integrity in GC, even for perturbations much longer than 2.5s (*Maier et al., 2015*; *Flores et al., 2018*); furthermore, the same rats in later sessions produced normally-timed orofacial responses on the control trials. We suggest that, to at least some degree, such effects on behavior reflect the widespread nature of taste processing, and the status of GC as one participatory node.

Despite being just one node of this large network of brain regions, our brief perturbations reveal a temporally-specific role of GC in the driving of orofacial behavior - a role that could not be discerned through wholesale disruption of activity. This conclusion is bolstered by findings showing that: (1) even early - that is pre-transition - GC perturbations delay gaping; and (2) palatability-related firing does not immediately return to normal levels following cessation of perturbation (as would be expected if GC was simply an output path reflecting processing performed elsewhere). Our 0.5s perturbations reveal that GC contributes to the instigation of a gaping bout but plays no role in the maintenance of gaping once it begins. These data suggest a dynamic flow of processing control within the larger taste network: modulatory signals propagate out of GC (signals that likely develop under the guidance of basolateral amygdala; *Piette et al., 2012*) to influence the choice of a motor program in brainstem circuits, which is then implemented and controlled locally.

At its heart, the proposed role of cortex in this model of taste processing has deep similarities to the role of neuromodulatory systems in the circuits underlying *Aplysia* feeding (*Dacks and Weiss, 2013*), leech swimming (*Crisp and Mesce, 2004*), control of gastric rhythms in the lobster and crab (*Marder and Bucher, 2007*), and rat whisking (*Hattox et al., 2003*). In each of these sensorimotor

systems, rhythmic motor programs are produced autonomously by multi-functional pattern generating circuits – that is, the patterns can be observed even when the pattern generators are isolated from most of the rest of the nervous system. In situ, however, the specific rhythms produced by the pattern generators are influenced by descending 'modulatory' signals of a neural or hormonal nature. We argue that the post-transition output of GC, while a neural code in its own right, is just such a modulatory influence on the brainstem central pattern generator for orofacial behaviors (see also below).

The discreteness, coherence and inter-trial variability of the ensemble dynamics suggests that GC functions as an attractor network in playing this role (*Hopfield, 1982*; *Amit, 1992*): (1) attractor networks with multiple quasi-stable states can reproduce the sudden switches of activity seen in GC ensembles (*Miller and Katz, 2010*); (2) the transition durations and state lifetime statistics of GC population dynamics are more in line with a dynamically switching attractor model than linear models of firing rate evolution (*Jones et al., 2007*; *Sadacca et al., 2016*); and (3) nonlinear attractor-based circuits that exploit the noise inherent in neural processing more optimally perform the decision to ingest or expel a taste, which rats need no training to perform, than do linear integrating circuits (*Miller and Katz, 2013*). Our optogenetic protocol, with its mix of inhibitory and excitatory effects, presumably introduces a transient disruption in such attractor dynamics; such a perturbation is strong enough to transiently 'knock' the network out of stability, but only if it hasn't already settled into the eventual, decision-related stable state.

Our data demonstrate that perturbations beginning 50ms after ensemble neural transitions to palatability-related firing have no impact on gaping. Given that gaping lags 0.2–0.3s behind this transition (*Sadacca et al., 2016*), it appears that GC becomes irrelevant to gaping even before actual gaping has begun. Inherent limitations of change-point modeling in 0.5s perturbation trials made it impossible to directly test smaller intervals than 50ms, but our analyses suggest that it is likely that gaping was unperturbed by middle 0.5s perturbations even in trials in which transitions occurred between 0.65 and 0.7s. While one could hypothesize reasons that perturbation beginning very soon after an ensemble transition should be effective in delaying gaping – the fact that transitions between quasi-stable states of GC processing, however sudden, cannot be instantaneous (time constants of neural firing ensure that there is some finite, albeit small amount of time across which the ensemble makes the 'jump' from one state of activity to another), for instance, and the fact that our change-point analysis technique provides only a noisy estimate of state transition times – our data and analyses suggest that GC perturbation becomes ineffective at delaying gaping almost immediately following the transition. GC appears to emit a 'gape signal', after which it is no longer involved in the implementation of that signal.

In this study, we focused exclusively on gapes, the orofacial responses that rats make to expel aversive tastes from the oral cavity (pilot attempts to implant EMG electrodes in deeper muscles that control the distinctive consumption behaviors that occur in response to palatable tastes resulted in unacceptable levels of distress for the animals). This means that it remains possible that gapes and 'palatable taste acceptance behaviors' are produced by separate cortical mechanisms (*Peng et al., 2015*), and that therefore our results are informative only about aversion. We consider this possibility highly unlikely, however, for several reasons: (1) GC ensemble firing reflects the palatability of both appetitive and aversive tastes (*Figure 3*, *Katz et al., 2001*; also see *Fonseca et al., 2018*), even if palatability is modified by learning (*Moran and Katz, 2014*); (2) the latency and inter-trial variability of the onset of palatability-related ensemble activity is similar for palatable and aversive tastes (*Sadacca et al., 2016*); (3) there is considerable overlap in the brainstem circuits that underlie the distinct palatability-related orofacial behaviors (*Travers et al., 2000*; *Chen and Travers, 2003*; *Venugopal et al., 2007*; *Moore et al., 2014*), resulting in similar latencies in the onset of these behaviors after taste delivery (*Travers and Norgren, 1986*); and (4) independent analysis has suggested that orofacial behaviors reflecting aversiveness and palatableness lie on a single parametric continuum (*Breslin et al., 1992*). These lines of evidence are consistent with the suggestion that cortex plays similar roles in the initiation of acceptance and rejection behaviors, which leads us to speculate that the transition of GC population activity to reflect stimulus palatability marks a shift in processing control, irrespective of the palatability of the tastant.

In summary, the balance of our results demonstrate a dynamic role for cortex in the processing of tastes; because this role involves ensemble activity states with variable trial-to-trial latencies, it cannot be discerned using standard analyses that average across trials – trial-specific analyses, such as

those underlying the results in *Figure 7*, supersede across-trial analyses such as those described in *Figure 3*. They reveal the importance of a unique moment in time that, despite being massively variable in latency from trial to trial, denotes a reliable shift of processing control - a modulatory signal emerging (at least partly) from cortical circuits that is passed (presumably) to a brainstem central pattern generator. These results suggest an attractor-like network of activity (although they could also be consistent with networks with thresholds), potentially spread across interconnected brain regions, underlying the animal's decision to ingest or expel the tastant in the mouth - perturbations to this network can disrupt its functioning transiently, but only if it has not yet settled into the final, behaviorally-relevant stable state.

## Materials and methods

### Experimental design

#### Subjects

Adult, female Long-Evans rats (n = 5; 275–300g at time of virus injection; 300–350g at time of electrode implantation) served as subjects in our study (in our hands, female Long-Evans rats have proven more docile than males, but we have observed no sex differences in the basic cortical dynamics of taste responding). The rats were housed in individual cages in a temperature and humidity controlled environment under a 12:12 hr light:dark cycle. All rats were given *ad libitum* access to food and water before experiments started. Rats were weighed daily and observed to never drop below 80% of their pre-surgery weight. All experimental methods were in compliance with National Institutes of Health guidelines and were approved in advance by the Brandeis University Institutional Animal Care and Use Committee.

We also performed a set of control analyses on data taken from 10 adult, female Long-Evans rats, previously published in *Sadacca et al. (2016)* and *Li et al., 2016*.

#### Virus injections

We injected adeno-associated virus (AAV9) coding for ArchT and green fluorescent protein (AAV9-CAG-ArchT-GFP, $2.5 \times 10^{11}$ particles per mL) into GC. This AAV serotype has been shown to effectively spread to and infect all cell types (*Aschauer et al., 2013*) in regions including GC (*Maier et al., 2015*; *Li et al., 2016*). As our critical tests involved within-session comparisons of different inactivation timings, there was no need for a separate group of control rats injected with virus coding for GFP alone (see below).

Rats were first anesthetized using a ketamine/xylazine mixture (1mL ketamine, 0.05mL xylazine/kg body weight) delivered *via* an intra-peritoneal injection. Supplemental anesthetic injections were given as needed. The head was shaved, cleaned with an iodine solution and 70% ethanol, and positioned into the stereotax. We then excised the scalp and cleaned and leveled the top of the skull. Small craniotomies were drilled bilaterally over the location of GC (anteroposterior +1.4mm from bregma, mediolateral ±5mm from bregma; *Paxinos and Watson, 2007*), the meningeal tissues were gently excised, and virus was infused.

We lowered a glass micro-pipette (tip diameter: 10–20μm) filled with the infusate (virus particles suspended in a solution of phosphate-buffered saline (PBS) and Oregon Green 488 (Invitrogen) into the centers of the craniotomies, and performed a sequence of 3 injections bilaterally into GC: at 4.9, 4.7 and 4.5mm ventral to dura, virus was injected in discrete pulses (44 pulses/location, with 25nL per pulse, 7s between consecutive pulses = 1.1μL total volume injected per depth) controlled by a Nanoject III microinjector (Drummond Scientific). Following each unilateral set of injections, the micropipette remained in place for 5 min, after which it was smoothly removed over the course of 1 min so that fluid would not spread back up the micro-pipette track. Craniotomies were then sealed with silicone (Kwik-Sil, WPI), the scalp was sutured, and the rat was given analgesic (meloxicam 0.04mg/kg), saline and antibiotic (Pro-Pen-G 150,000 U/kg) injections. Similar antibiotic and analgesic injections were delivered 24 and 48 hr later.

Rats were allowed to recover for 4–6 weeks from this procedure, in order to ensure adequate infection and subsequent expression of optical channels (ArchT) and GFP.

## Opto-trode, intra-oral cannula and EMG electrode implantation

After recovery from virus infusion surgery, rats were again anesthetized, and implanted with bilateral GC opto-trode bundles. Each bundle consisted of either 30 or 32 recording microwires (0.0015inch formvar-coated nichrome wire; AM Systems) and one optical fiber (0.22 numerical aperture, 200µm core, inserted through a 2.5mm multimode stainless-steel ferrule; Thorlabs). The microwire bundle was glued to a custom-made electrode-interface board (San Francisco Circuits) and connected to a 32 channel Omnetics connector. In the case of the 30 microwire bundles, the final two pins were connected to two electromyography (EMG) electrodes (PFA-coated stainless steel wire; AM Systems) implanted into the digastric muscle under the jaw. Finally, the microwires and optical fiber were connected to a custom-built 3D printed microdrive that allowed the entire assembly to be moved ventrally after implantation. The microwire tips were located 0.5mm ventral to the tip of the optical fiber - this maximized the likelihood that the electrodes recorded the activity of neurons that were illuminated by the laser. For more information on the implanted apparati and associated electronics, see *Katz et al. (2001)*, *Sadacca et al. (2016)* and *Li et al. (2016)*, as well as the Katz Lab webpage.

Rats were anesthetized, after which we shaved and cleaned the scalp and situated the head in the stereotax. After excising the scalp and leveling the skull, we drilled five self-tapping screws into the skull for supporting and grounding the opto-trode bundles. The silicone seal was removed from the craniotomies, as were any tissues that had grown in since the prior surgery. We then slowly (over 5–10 min) lowered the opto-trode bundles to a depth of 4.3mm from the dura mater (0.2mm above the most dorsal location of virus injection). The ground wires were wound tightly around the skull screws and the bundles were cemented in place with dental acrylic. The optical fiber was looped so that the ferrule could be cemented away from the microdrive - this configuration reduced the stress on the microdrive when the animal was later plugged in to the experimental apparatus.

Once the opto-trode assembly was cemented in place, the rat was removed from the stereotax and implanted with a single (right-side) intra-oral cannula (IOC) for controlled delivery of tastants on the tongue. IOCs were made with thin polyethylene tubing and inserted in the space between the first maxillary molar and the lip, through the masseter muscle and inside the zygomatic arch, and out through the opening in the scalp (*Phillips and Norgren, 1970*; *Katz et al., 2001*) The IOC was topped with a plastic connector that could be attached to the taste delivery apparatus, and cemented in place with dental acrylic.

The EMG electrodes were channeled down the left side of the face (opposite from the IOC); after the overlying skin had been teased away from the belly of the digastric muscle, one end of each EMG electrode was tied to a suture needle, which was then inserted into the muscle, such that the electrode could be pulled into the desired position (for more details, see *Loeb and Gans, 1986*; *Travers and Norgren, 1986*; *Dinardo and Travers, 1994*; *Li et al., 2016*). The electrode wires were trimmed and held in place with vetbond tissue adhesive (3M) and the skin covering the anterior digastric was sutured back into place. Finally, a modified falcon tube was glued to the front of the headcap as a protective cap, and bacitracin ointment was applied all around the base of the headcap and over the wound under the jaw.

Rats were postoperatively injected with analgesic (Buprenophine 0.05mg/kg), saline, and antibiotic (Pro-Pen-G 150,000 U/kg). Similar antibiotic, saline and analgesic injections were delivered 24, 48 and 72 hr later, and bacitracin ointment was reapplied. The rats were handled every day and allowed to recover to 90% of their pre-surgery weight (at least 7 days after surgery) before being introduced to the experimental apparatus.

## Habituation

Following recovery from the opto-trode implantation surgery, we habituated rats to passive water deliveries for 3 days before beginning data collection. In these daily habituation sessions, we attached the rats to the electrophysiology acquisition system, laser patch cables and taste delivery apparatus, and infused 100 pulses of distilled water (~40µL per pulse; 15s inter-pulse interval) into the animal's oral cavity through the IOC. Starting with the second habituation day, we also placed rats on a mild water restriction schedule - 20mL of water (not including the 4mL delivered during habituation sessions themselves) per day. This water restriction schedule was maintained for the duration of the study (~7 days per animal).

Opto-trode bundles were driven deeper after each habituation session using the microdrive built into the assembly; by the end of the habituation period, the distance traveled was 0.2mm, such that the tips of the electrodes lay within the region of GC infected with the virus.

## Passive taste administration and laser stimulus delivery

We used 2 concentrations of palatable sucrose (30mM: Dilute Sucrose (Dil Suc), 300mM: Concentrated Sucrose (Conc Suc) and of aversive quinine-HCl (0.1mM: Dilute Quinine-HCl (Dil Qui), 1mM: Concentrated Quinine-HCl (Conc Qui) dissolved in distilled water as the stimuli in our experiments. Concentrated sucrose and quinine evoke strongly valenced orofacial responses; the dilute stimuli are of similar but far less extreme palatability – a fact that aided in the analysis of palatability-related neural firing (*Li et al., 2016*; see also below). The taste delivery apparatus consisted of gently pressurized tubes containing taste solutions; the tubes converged upon a manifold of finer polyamide tubes that could be inserted into (to 0.5mm past the end of) the IOC, thus eliminating any chance of mixing. The manifold could be locked securely into the dental acrylic cap. The tastes were then delivered under slight nitrogen pressure - this taste delivery protocol has been consistently shown to ensure reliable tongue coverage at short latencies (*Katz et al., 2001*; *Sadacca et al., 2016*; *Li et al., 2016*).

Data were collected during sessions in which GC was briefly perturbed during tasting trials. In the majority of these trials ('0.5s perturbation' trials), the laser was turned on for 0.5s at either 0.0, 0.7, or 1.4s post taste delivery ('early', 'middle' and 'late' 0.5s perturbation trials; 25% of the trials each); the remaining 25% of the trials were no-laser controls. Note that all critical tests were comparisons of the impacts of different trial types within single sessions (whenever 'control trials' are mentioned, we will be referring to no-laser trials from the same session), a fact that obviated the need for 'empty virus' control rats: we are not interested in a simple laser vs no-laser difference, but rather are asking whether two identical (but differently timed) perturbations have different impacts. All effects were differences observed between trial types in the same rat, which means that they necessarily were not the result of general, confounding impacts of particular treatments – they were by definition differences between different timings of neural perturbation, evinced under identical conditions. Nonetheless, we did (for completeness sake) give a subset of rats a second session in which 50% of the trials involved full-trial (2.5s) perturbations (and the other 50% were no-laser control trials). One experimental session was run per day; when rats were run for two or more sessions, we counterbalanced session type – that is, $\frac{1}{2}$ of these rats experienced 2.5s perturbations in the first session and 0.5s perturbations the following day, and $\frac{1}{2}$ *vice versa* (see below).

Sessions with 2.5s perturbations consisted of 8 sets of trials (two sets per taste - one with the lasers on and one with no laser). Each set included 15 trials, for a total of 120 trials per session. Similarly, sessions with 0.5s perturbations included 16 sets of trials (four sets per taste - early, middle, and late 0.5 perturbations, and no-laser control trials). To keep the total number of trials per session from ballooning (a basic concern in taste research is the awake animal's finite appetite), each set included only eight trials (total, 128 trials per session). Again, we moved the opto-trode bundle 0.075mm ventrally (deeper into GC) prior to each session, to ensure that we obtained fresh units in every session. Trials were delivered in pseudo-random order and each involved delivery of ~40μL of fluid through the IOC, for a total volume of 5mL per session.

We used a 532 nm, DPSS laser (Laserglow Technologies), connected to the implanted ferrules using standard FC/PC patch cables (Thorlabs), for all optogenetic perturbations. Taste and laser delivery were controlled through a Raspberry Pi computer. The strength of the laser input was calibrated, prior to opto-trode implantation, to yield an illumination power of 40mW at the tip of the optical fiber. This output power perturbs all ArchT infected neurons in a $1mm^3$ sphere below the tip of the fiber *in vivo* (*Han et al., 2011*; *Yizhar et al., 2011*) - a sphere that encompasses about 33% of GC in the caudal-rostral axis (*Kosar et al., 1986*; *Maier et al., 2015*; *Li et al., 2016*). These parameters have previously been shown to reduce the activity of ArchT+ cortical neurons with minimal latency and damage (*Maier et al., 2015*; *Li et al., 2016*; *Flores et al., 2018*).

## Acquisition of electrophysiological data

We collected 30,000 voltage samples per second from each implanted neural and EMG electrode, using a 32-channel analog-to-digital converter chip (RHD2132) from Intan Technologies. These chips

are capable of recording voltage signals over a wide range of frequencies (0.1Hz-20kHz) and amplitudes (microvolts to millivolts), thereby enabling us to record neural and EMG signals through the same hardware system. The experimental chamber was ensconced in a Faraday cage that shielded recordings from external electrostatic and electromagnetic influences.

## Histology and evaluation of GFP expression

In preparation for histology, rats were deeply anesthetized with an overdose of the ketamine/xylazine mixture, after which DC current (7μA for 7s) was passed through selected microwires, marking the area below the electrode tips. We perfused the rats through the heart with 0.9% saline followed by 10% formalin and harvested the brain. The brain tissue was incubated in a fixing mixture of 30% sucrose and 10% formalin for 7 days before GC was sectioned into 50μm coronal slices.

We rinsed the slices 3 times with 1X-PBS over 15 min and permeabilized them in a 0.3% Triton X-100 + 1% normal Donkey serum + 1X-PBS blocking solution for 2 hr at room temperature. We replaced the blocking solution with primary antibody solution (1:500 anti-GFP-rabbit IgG; Life Technologies) for 12 hr at 4°C. After incubation with the primary antibody, the slices were rinsed with 1X-PBS 3 times over 15 min followed by incubation with the secondary antibody incubation of (1:200 Alexa Flour 488 donkey anti-rabbit IgG (H + L); Life Technologies) for 12 hr at 4°C. After a final set of rinses with 1X-PBS (3 times over 15 min), we mounted the slices on charged glass slides and coverslipped them with Fluoromount Aqueous Mounting Medium. Slices were imaged with a Keyence fluorescence microscope to confirm successful virus infection and opto-trode location for each animal.

The spread of AAV in GC was evaluated *via* the expression of GFP, as has been done previously (*Maier et al., 2015*; *Li et al., 2016*; *Flores et al., 2018*).

## Data analysis

Most statistical analyses in this paper were performed using Bayesian methods implemented in the PyMC3 probabilistic programming package (*Salvatier et al., 2016*). Although the far more common practice in the literature is to implement analyses similar to ours in a frequentist/maximum likelihood estimation (MLE) paradigm, the Bayesian approach offers several advantages. For one, Bayesian statistics provides a natural way to infer the entire joint posterior distribution of the model parameters in the light of the data at hand. This allows the Bayesian methodology to make robust inferences without being constrained by the sampling-related assumptions of parametric frequentist statistics or the lack of statistical power of non-parametric frequentist techniques. Relatedly, the flexibility of the Bayesian framework allows the construction of statistical models appropriate for the data-generating process that can include non-standard (such as multi-modal) parameter distributions. Such models (of which we use several in this study) often cannot be accommodated by frequentist approaches at all, even if they are 'true' descriptions of the underlying generative process. Finally, despite working with highly flexible models, Bayesian approaches provide the added advantage of using model priors to regularize parameter estimates - we use 'weakly informative' priors in our analyses that are known to reduce the susceptibility of the inference process to noise by penalizing model flexibility (unless supported by the observed data). (For a detailed comparison of frequentist and Bayesian estimation in statistics and a discussion of weakly informative priors, please refer to *Gelman et al., 2013* and *McElreath, 2015*). We will describe the properties of each statistical model used in our analyses, and our specific prediction(s) for each such model, in the sub-sections below.

Recent advances in statistical computing have made it possible to circumvent the analytical challenges that have historically plagued the application of Bayesian techniques to many practical problems. In particular, new Markov Chain Monte Carlo (MCMC) techniques have been developed to facilitate arriving at an approximation to the posterior distribution of the model parameters by drawing samples from it. We performed inference in our Bayesian probabilistic models using the No-U-Turn-Sampler (NUTS; *Hoffman and Gelman, 2014*), a state-of-the-art, self-tuning Hamiltonian MCMC algorithm that efficiently draws samples from the posterior distribution described by the data at hand. The performance of the sampler can be evaluated by running several independent sampling chains - a properly tuned sampler that explores the parameter space in an unbiased manner and draws samples from the correct posterior distribution will result in all the chains 'converging' to the same distribution. Statistically, this is evaluated by computing the Gelman-Rubin $\hat{R}$ statistic

(*Gelman and Shirley, 2011*) across all the sampling chains. $\hat{R}$ close to 1 indicates that the sampling runs have converged and produced samples from the same posterior distribution (we allow values from 0.99 to 1.01). Each analysis finally reports the uncertainty for the inferred parameters as 95% credible intervals - essentially the interval that covers 95% of the probability mass under the posterior distribution of the parameters. Credible intervals inherently serve as significance tests in this setting - for instance, if the 95% credible interval for an estimated parameter does not overlap 0, we can conclude that this parameter is different from 0 at the 5% level of significance.

## Single unit isolation

We followed a semi-supervised spike sorting strategy: intra-cranial voltage data was filtered between 300–3000Hz, and a Gaussian Mixture Model (GMM) identified potential clusters which were refined manually. For more details on our spike sorting methods and its efficacy in isolating single units, please consult *Mukherjee et al. (2017)*. Our spike sorting code is freely available at https://github.com/narendramukherjee/blech_clust (*Mukherjee, 2019*; copy archived at https://github.com/elifesciences-publications/blech_clust).

## Impact of optogenetics on neural firing

We built a hierarchical Poisson generalized linear model (GLM) for the spiking of a single neuron to evaluate the impact of optogenetic perturbations on firing. Hierarchical GLMs provide precise estimates of condition-specific model parameters, especially when they are expected to vary around condition-agnostic means. In our case, the model parameters are the mean firing rates for every taste and optogenetic perturbation condition, that are in turn composed of taste- and perturbation-specific effects ('random effects') and means across tastes and perturbation conditions ('fixed effects'). Coupled with the Poisson distribution's suitability for count (here spikes) data, this model can accurately estimate the change in neural firing induced by perturbations.

For each neuron $n$ in our dataset, we aggregated the spikes produced on trial $i$ of taste $T$ in optogenetic perturbation condition $O$. There were four levels for $T$ corresponding to the tastes in our dataset: Dil Suc, Conc Suc, Dil Qui and Conc Qui. The number of levels for $O$ depended on the type of optogenetic perturbation being delivered in the session: in the 2.5s perturbation sessions, $O$ had two levels, corresponding to the laser off (control) and on trials respectively; the 0.5s perturbation sessions had 3 types of perturbation trials - starting at 0s (early 0.5s), 0.7s (middle 0.5s) or 1.4s (late 0.5s) after taste delivery - and therefore had six levels for $O$ (a 'laser off-laser on' pair for each of the 3 types of perturbations). Our model posits that the aggregate number of spikes $S_{n,i,T,O}$ is Poisson-distributed with a mean ($firing_{n,T,O}$) that depends on the taste ($\mu_T$), optogenetic perturbation ($\mu_O$) and an interaction between the taste and optogenetic perturbation ($\mu_{T,O}$). As described above, owing to the hierarchical structure of the model, each of these effects is further composed of a fixed effect and a random effect. Using weakly informative Gaussian and Half-Cauchy priors for the mean and variance parameters respectively, our model formally says:

$$
\begin{aligned}
\text{Fixed effects} &: F_1, F_2, F_3 \sim \mathcal{N}(0, 10) \\
\text{Variances} &: \sigma_1, \sigma_2, \sigma_3 \sim Half\ Cauchy(1) \\
\text{Taste} - \text{specific means} &: \mu_T \sim \mathcal{N}(F_1, \sigma_1) \\
\text{Optogenetics} - \text{specific means} &: \mu_O \sim \mathcal{N}(F_2, \sigma_2) \\
\text{Taste} - \text{and} - \text{optogenetics} - \text{specific means} &: \mu_{T,O} \sim \mathcal{N}(F_3, \sigma_3) \\
\text{Mean firing rate(with log link)} &: log(firing_{n,T,O}) = \mu_T + \mu_O + \mu_{T,O} \\
\text{Observed number of spikes} &: S_{n,i,T,O} \sim Poisson(firing_{n,T,O})
\end{aligned}
\tag{1}
$$

As explained in the introduction to the data analysis section, we used MCMC (specifically the NUTS sampler) to sample the posterior distribution of $firing_{n,T,O}$ for every taste and optogenetic perturbation. We performed this analysis for every neuron in our dataset and finally calculated the impact of perturbation on firing as the difference in $firing_{n,T,O}$ between no-laser (within-session control) and their corresponding perturbation trials. If the 95% Bayesian credible interval for these differences in $firing_{n,T,O}$ for a neuron did not overlap 0, we concluded that the perturbation significantly impacted the firing of this neuron (see the introduction to the data analysis section for a discussion of how Bayesian credible intervals inherently serve as significance tests).

## Regression of single neuron firing with palatability ranks

We analyzed, as we have done previously (*Sadacca et al., 2016*), the time course of palatability-related information in the activity of single neurons by regressing their firing rates on the palatability ranks of the tastes (Dil Suc: 3, Conc Suc:4, Dil Qui: 2, Conc Qui: 1; higher is more palatable). In order to estimate the firing rates of neurons, we aggregated the spikes of each neuron, on a trial-by-trial basis, in 250ms bins moved by 25ms steps. We divided the aggregate number of spikes by the width of the bins (250ms) to obtain the near-instantaneous firing rate of each neuron across time on individual trials.

These firing rates, of course, vary widely between neurons. Furthermore, correlations between firing rate and palatability ranks may be significantly positive (stronger firing to more palatable tastes) or significantly negative (stronger firing to more aversive tastes). We therefore needed to perform a 2-stage transform on neural firing before we could analyze all neurons as a group in our regression analysis. The first step was standardization - we transformed the firing rate of each neuron in each time bin by subtracting the trial-averaged firing rate of the neuron in that time bin and scaling by its standard deviation across trials (to get z-scores), ensuring that the firing rates of all neurons were on a comparable scale. Next, we multiplied the standardized firing rate of each neuron by the sign of the time-averaged Spearman rank correlation coefficient between its firing and the palatability ranks. This ensured that the sign of the relationship of neural firing with palatability was the same for all neurons in our dataset, but left the magnitude of that relationship unaffected.

Our statistical model treats the standardized firing rate $firing_{t,P,i}$ of a neuron at time bin $t$ on trial $i$ of a taste with palatability rank $P$ as Gaussian-distributed with a mean $\mu_{t,P}$ that depends linearly on $P$. We defined the palatability index in time bin $t$, $\beta_{Palatability,t}$, as the change in $\mu_{t,P}$ induced by a unit change in $P$. $\beta_{Palatability,t}$ is, therefore, the slope of the line that explains $\mu_{t,P}$ in terms of $P$, an estimate of the strength of the firing-palatability relationship. Using weakly informative Gaussian and Half-Cauchy priors for the mean and variance parameters respectively, our model formally says:

$$
\begin{aligned}
\text{Prior on palatability index} &: \beta_{Palatability,t} \sim \mathcal{N}(0,1) \\
\text{Prior on observation noise} &: \sigma \sim Half\ Cauchy(1) \\
\text{Mean firing rate} &: \mu_{t,P} = \beta_{Palatability,t} \times P \\
\text{Firing rate} &: firing_{t,P,i} \sim \mathcal{N}(\mu_{t,P}, \sigma)
\end{aligned}
\tag{2}
$$

We used MCMC to infer the posterior distribution of $\beta_{Palatability,t}$ across all neurons in our dataset (again, see above). The firing rate transformations defined previously put the activity of all neurons on the same scale and allowed us to infer a single posterior distribution of $\beta_{Palatability,t}$ across all the neurons in our dataset. We repeated this regression for each time bin $t$ from 0.25s before to 1.5s after taste delivery, obtaining posterior estimates of $\beta_{Palatability,t}$ specific to each time bin. Finally, we normalized $\beta_{Palatability,t}$ by subtracting its average baseline value (from 0.25 to 0s before tastes). We report the baseline-normalized $\beta_{Palatability,t}$ as the palatability index $\beta_{Palatability}$.

## Characterizing the time course of the palatability index

In a manner similar to our previous work (*Sadacca et al., 2016*), we modeled the time course of the posterior mean of the single neuron palatability firing index, $\overline{\beta}_{Palatability}$, with a logistic sigmoid. The difference between the lower and upper asymptotes of the S-shaped logistic function fits the total rise in $\overline{\beta}_{Palatability}$ across time, while its slope describes the rate of this rise. As $\beta_{Palatability}$ was already normalized to its average pre-stimulus value, we set the lower asymptote of the logistic function to 0. With weakly informative Gaussian priors (restricted to positive values) on the upper asymptote ($L$), slope ($k$) and inflection time ($t_0$, ms post taste delivery) of the logistic sigmoid, our model is as follows:

$$\text{Prior on upper asymptote} \quad :L \sim \begin{cases} \mathcal{N}(0,0.1) & L>0 \\ 0 & \text{otherwise} \end{cases}$$

$$\text{Prior on slope} \quad :k \sim \begin{cases} \mathcal{N}(1,1.0) & k>0 \\ 0 & \text{otherwise} \end{cases}$$

$$\text{Prior on inflection time} \quad :t_0 \sim \begin{cases} \mathcal{N}(675ms,75ms) & t_0>0 \\ 0 & \text{otherwise} \end{cases} \tag{3}$$

$$\text{Prior on observation noise} \quad :\sigma \sim Half\ Cauchy(1)$$

$$\text{Mean palatability index} \quad :\overline{\beta}_{Palatability}(t) \sim \mathcal{N}(\frac{L}{1+e^{-k(t-t_0)}},\sigma)$$

We defined the peak of the palatability firing index, $t_{peak}$, as the time (post taste delivery) when $\overline{\beta}_{Palatability}$ reached 95% of its maximum value, $L$. We transformed the posterior distributions of $L$, $k$ and $t_0$ to get $t_{peak}$ (inferred using MCMC) as follows:

$$t_{peak} = \frac{\ln\frac{95}{5}}{k} + t_0 = \frac{\ln 19}{k} + t_0 \tag{4}$$

We performed this analysis on aggregates of control trials, and for comparison we performed the identical analysis separately on early, middle and late 0.5s perturbation trials. Note, however, that these latter analyses can only be interpreted with great caution, because they violate the basic stationarity assumption of curve fitting – the analysis assumes that all of the data from which the function is to be fit were pulled from the same conditions, and thus they may reasonably be thought to reflect a single function; the sudden onset of the laser-driven perturbation necessarily alters those conditions, as does the sudden end of that perturbation. In addition, the standard practice of using Gaussian-distributed observation noise in curve fitting (both in our Bayesian analysis and in least-squares based approaches) leaves the fitted function vulnerable to outliers that are bound to emerge if the data are non-stationary; for instance, the late 0.5s perturbations impact $\overline{\beta}_{Palatability}$ well after its peak (*Figure 3A3*) - but trying to fit $\overline{\beta}_{Palatability}$ in this condition with a single sigmoid affects both $t_{peak}$ and $L$, features of the fitted sigmoid that are attained earlier in time (*Figure 3C*). The control (no-laser) trials as well as the 2.5s perturbation trials do not have laser onset/offset during the 1.5s window subjected to the palatability analysis and thus, do not violate this assumption.

The slopes of these fits are thus probably untrustworthy; in fact, our results and previously published work (*Sadacca et al., 2016*) demonstrate that even the slope of control data is an artifact of across-trial aggregation, and unrepresentative of single-trial data. However, despite the fitted function's susceptibility to outliers, the height ($L$) and latency ($t_{peak}$) of the asymptote of the sigmoid are features that can be reasonably interpreted for the majority of the 0.5s perturbation conditions, and we focus upon these properties in the Results and in *Figure 3*.

## Modeling and change-point identification in ensemble firing data

As described in the Introduction, previous analyses reveal that rat GC population activity in response to a taste consists of a sequence of 3 coherent, abruptly-appearing ensemble states (*Katz et al., 2001*; *Jones et al., 2007*; *Sadacca et al., 2012*; *Sadacca et al., 2016*; *Li et al., 2016*) in which firing rates 'code', in turn, taste presence, taste identity, and taste palatability; the transition into this last state has particular relevance for the prediction of palatability-related behavior in single trials, and is the subject of this study. While identifying these sequences typically requires several forward and backward passes through a dataset made up of many identical (i.e., unperturbed) trials, the work already published on the nature of these state sequences (see also *Jones et al., 2007* and *Moran and Katz, 2014*) renders it possible (for the purposes of the current study) to simplify the analysis by more concretely defining this process as involving ensemble firing change points between states having the following properties (also see *Figure 6*):

1. **Detection state:** a single distribution of population activity for all the tastes, indicating taste presence on the tongue.
2. **Identity state:** two distinct distributions of population activity, for the two taste identities in our experiments (Suc and Qui).

3. **Palatability state:** four distinct distributions of population activity, for the four taste palatabilities in our experiments (Dil Suc, Conc Suc, Dil Qui and Conc Qui).

With this characterization we were able to design a relatively simple change-point model that allowed us to detect these coherent transitions in population activity in individual trials. We first prepared the data for the change-point model by aggregating the spikes of each neuron in each trial into 10ms non-overlapping bins, indexing each neuron recorded in a session with a scalar $i$ running from 0 to the number of neurons in the session $N$. We then converted the aggregate spiking data to a categorical format by marking each time bin by the index $S$ of the neuron that spiked in that bin, with $S = 0$ corresponding to no spikes from any neuron. If more than one neuron spiked in a time bin - a highly uncommon occurrence, given the relatively low firing rates of GC neurons and the small (10ms) bins being used - we randomly selected one of the spiking neurons for assignment to that bin (*Jones et al., 2007*; *Sadacca et al., 2016*).

With the (processed) categorical spiking data in hand, we now designed the change-point model to describe the ensemble firing in each of the three states (listed above) as categorical distributions with $N + 1$ emissions, with 1, 2 and 4 such distributions corresponding to the detection, identity and palatability states respectively. Note that the results of this analysis are unchanged if we relax the parameters slightly to allow for 4 'state 2' distributions – that is, if we allow the Identity State to differ for the different concentrations of Sucrose and Quinine; this is probably because while many neurons may code different NaCl concentrations distinctly (*Sadacca et al., 2012*), for other tastes the vast majority of neurons appear to code quality rather than concentration (see, for instance, *Fonseca et al., 2018*).

We analyzed 1.5s of ensemble activity post taste delivery from each of the four 0.5s perturbation conditions in the 0.5s perturbation sessions. For the control (no-laser) trials, this corresponded to 0.0–1.5s of firing after taste delivery. On the 0.5s perturbation trials, we ignored the 0.5s during which the lasers were on - for example, we analyzed 0.5–2.0s of firing post tastes when the lasers were on from 0 to 0.5s. In the resultant 1.5s of activity, we assumed that the change from detection to the identity state, $C_I$, happens anywhere in the interval [0.2s, 0.6s] (except in the early 0.5s perturbation trials, where we allowed the identity state to start earlier from 0.1s, to account for the possibility that some amount of taste processing happens in GC even while the neurons are being perturbed). The second change point, $C_P$, from identity to palatability firing, was assumed to occur anywhere in the interval [$C_I$ + 0.2s, 1.3s] (except in the middle 0.5s perturbation trials, where the palatability state can start earlier at $C_I$ + 0.1s for the same reason). Thus, we place uniform priors over the intervals that define $C_I$ and $C_P$, corresponding to the timing of sudden, coherent firing rate transitions in GC ensembles (*Jones et al., 2007*; *Sadacca et al., 2016*).

While we used MCMC sampling for inference in all the statistical analyses described above, we did not do so in the change-point model. The reason for this decision is straightforward: given our specification of 2 change-points, the posterior distribution of $C_I$ and $C_P$ turns out to have two modes, corresponding to different (but equivalent) 'labels' for the change-points. One mode labels the first change-point as $C_I$ and the second as $C_P$, and the second switches the ordering of the change-points - both are valid solutions. In such situations, MCMC samplers 'jump' between the posterior modes and fail to converge on a solution - this has been referred to as the 'label switching' problem (*Jasra et al., 2005*).

We instead used the Expectation-Maximization (EM) algorithm, a commonly used approach for approximate posterior inference in models with latent variables. When properly initialized, EM converges to one of the modes of a symmetrical posterior and avoids the label switching problem. Classically, starting from a random initialization of the model parameters, the EM algorithm repeatedly cycles between finding the expectation (average) of the latent variables of the model ('E-step') and using these expected values to arrive at a better set of parameters by maximizing the model's likelihood function ('M-step') (*Bishop, 2016*). This process is guaranteed to increase the model's likelihood at every iteration; the algorithm ends once the improvement in the likelihood of the model tapers off.

For the process of inferring the parameters (firing rates) and latent variables (change points) of the change-point model, we slightly modified the E-step of the EM algorithm (we left the M-step unchanged) to directly pick the mode of the posterior over the latent variables (instead of its expectation). This was done to optimize computational speed and stability - using the full EM algorithm,

with the usual E-step (run on a subset of the data), did not change the inferred parameters significantly.

Our EM algorithm started with a randomly chosen set of initial emission probabilities $\alpha_D$, $\alpha_I$ and $\alpha_P$ for the categorical emissions that define the detection, identity and palatability states respectively. $C_I$ and $C_P$ are the latent variables of the model that control the timing of the transition of firing-rates from $\alpha_D$ to $\alpha_I$ and from $\alpha_I$ to $\alpha_P$. The algorithm then cycled between the following two steps:

1. **'Hard' E-step:** Pick the combination of the latent variables, $C_I$ and $C_P$, that has maximum posterior probability given the observed categorical spikes $S$ and the ensemble firing probabilities $\alpha_D$, $\alpha_I$ and $\alpha_P$. A standard E-step would make a 'soft' assignment at this point by averaging $C_I$ and $C_P$ over their respective posteriors. Instead, we make a 'hard' assignment by directly picking the mode of the joint posterior distribution of $C_I$ and $C_P$.

2. **M-step:** Set the categorical firing probabilities for each state to values that maximize the likelihood of the data given the $(C_I, C_P)$ pair picked in the E-step. This is proportional to the number of emissions of each neuron in that state. For example, with $S_t$ as the emission observed at time $t$, the likelihood-maximizing emission probabilities of neuron $n$ can be calculated as:

$$
\begin{aligned}
\text{In detection state} \quad &: \quad \alpha_{D,n} = \frac{\sum\limits_{t=1}^{C_I} \mathbb{1}_{(S_t=n)}}{\sum\limits_{n=1}^{N}\sum\limits_{t=1}^{t=C_I} \mathbb{1}_{(S_t=n)}} \\[2em]
\text{In identity state} \quad &: \quad \alpha_{I,n} = \frac{\sum\limits_{t=C_I}^{C_P} \mathbb{1}_{(S_t=n)}}{\sum\limits_{n=1}^{N}\sum\limits_{t=C_I}^{C_P} \mathbb{1}_{(S_t=n)}} \\[2em]
\text{In palatability state} \quad &: \quad \alpha_{P,n} = \frac{\sum\limits_{t=C_P}^{1.5s} \mathbb{1}_{(S_t=n)}}{\sum\limits_{n=1}^{N}\sum\limits_{t=C_P}^{1.5s} \mathbb{1}_{(S_t=n)}}
\end{aligned}
\tag{5}
$$

where $\mathbb{1}$ is the unit function that is one when $S_t = n$ and 0 otherwise.

In order to deal with the possibility that EM can get stuck at sub-optimal local maxima of log likelihood, we ran the algorithm from 100 different random initializations of the $\alpha$ parameters. We monitored the log likelihood of the data given the model parameters and ran the algorithm to a convergence threshold of $10^{-8}$ (or a maximum of 300 iterations). Finally, we picked the run with the maximum log likelihood at convergence and reported the change points (and their posterior probabilities given $S$ and $\alpha$) found on this run.

Of course, an inevitable result of performing such analyses on discontinuous data - such as trials in which 0.5s of spiking is missing because of perturbation, and the spiking on the two sides of this 0.5s are concatenated - is a certain number of artifactual change-points identified around the start or end of the perturbation time. That is, there is a relatively high likelihood that the analysis will identify spiking rates just before the onset of perturbation to be different from those just after. This issue is handled in the Results and Discussion sections.

## Measuring aversive orofacial behaviors (gapes)

Bitter (e.g., Quinine) tastes cause rats to produce an orofacial behavior known as 'gaping', the purpose of which is to maneuver the offending substances to the front of the mouth for egestion. As such, gapes index the fact that the neural processing of the bitter taste has (in a certain sense) reached completion - the rat has 'decided' that it does not want to ingest the taste. The occurrence of gapes can be measured in a number of ways, the most common of which is *via* human coding of video recordings - in the best of circumstances, gapes are readily visible as large yawn-like movements.

Of course, the best of circumstances often fail to occur in rats free to move and rear. This fact, and the difficulty of getting precise measures of gape onset time from a visual record, renders video coding of gapes suboptimal for our purposes. Much more objective and less noise-ridden is evaluation of jaw electromyography (EMG), in which individual gapes are recognizable as particularly large-amplitude and large-duration electrical bursts (*Figure 4A1–A2*). We have previously built a quadratic

classifier to detect these bursts in ongoing anterior digastric EMG signals, achieving 75% accuracy (*Li et al., 2016*).

Even this approach has somewhat troubling limitations, however, as its failure to reach close to 100% accuracy indicates. These limitations stem from the facts that: (1) not all high-amplitude jaw movements are gapes; and (2) gapes vary widely in amplitude, and in fact some are small enough to appear similar in size to many other mouth movements (see *Figure 4A1–A2*). In practice, both types of variability leave the classifier subject to false positives that must be somehow recognized and removed - the former most notably at the beginning of trials (when the taste hits the tongue, causing 1–2 relatively large-amplitude licks).

One solution to these problems involves making simultaneous recordings from multiple jaw muscles, but pilot experiments left us concerned that such drastic infiltration of the jaw can compromise normal movement, which would make interpreting our results difficult. Instead, we decided to take advantage of another, more robust feature of gaping: the fact that gapes occur in 4–6Hz 'bouts' of anterior digastric activity (*Travers and Norgren, 1986*; *Li et al., 2016*). While identifying gaping bouts as time periods during which this rhythmicity dominates the EMG signal is also imperfect - it is probabilistic and involves smoothing across time - it largely solves the problems described above.

We instantiated just such an procedure here, applying a Bayesian spectrum analysis that estimates the posterior probability that a 4–6Hz rhythm underlies a short time series of EMG activity (see below for technical details). By this analysis, the probability of gaping to any taste is modestly elevated at trial onset (because of the initial large-amplitude licks), but it quickly drops to effectively zero for Sucrose, which therefore contributes nothing to the overall calculation of when gaping begins. On Quinine trials, in contrast, the probability waxes and wanes appropriately with the occurrence of gape bouts (*Figure 4B1–B2*), rising precipitously and reliably just prior to the first gape (detected in a subset of data with both video recordings and the quadratic classifier, *Figure 4D*).

In important ways, this analysis is analogous to the method of divining palatability-relatedness of single-neuron firing described above and used in many previous studies (*Fontanini and Katz, 2006*; *Sadacca et al., 2012*; *Li et al., 2013*; *Sadacca et al., 2016*; *Li et al., 2016*) - the electrophysiological signal (in this case, the posterior probability of the range of gaping frequency in the EMG signal) varies (i.e., correlates) with the palatability of the proffered taste, and we average these correlations to ascertain the palatability-relatedness of the signal at each time point. Sucrose contributes no information to this signal (because rats do not gape to these sucrose concentrations), so the overall average gaping latency is equivalent to the difference between the time distributions of gaping probability to Dil and Conc Qui (see *Grill and Norgren, 1978a*; *Travers and Norgren, 1986*), which can be statistically assessed as the Kullback-Leibler (KL) divergence (again, see technical details below). Not only does this procedure reveal the onset of orofacial behaviors reflecting aversion, it pits the two Qui concentrations against each other to get rid of most of the nonspecific gape-like EMG activity (mentioned above) which is of similar magnitude on both Dil and Conc Qui trials and does not contribute to the gape onset calculation.

Unlike previously used methods, in which (usually) trials where gapes could not be reliably detected were removed from further analysis, this algorithm combines EMG data from all the trials available, thereby allowing us to avoid making statistical comparisons between conditions with very different sample sizes. At the cost of being unable to precisely detect the specific timing of later gapes in a bout, this procedure provides an estimate of the average timing of the first gape (both robust and reliable enough for the purposes of the within-session, between-condition analyses performed here).

## Bayesian spectrum analysis (BSA) of EMG recordings

As detailed previously, we recorded voltage signals from two unipolar EMG electrodes implanted in the anterior digastric muscle at 30kSamples/s. We used the difference in the voltage recorded by the two electrodes as the EMG signal - this procedure helps to cancel out any large artifacts produced by the animal's movements and is equivalent to using a differential amplifier (as done in *Li et al., 2016*). We down-sampled the EMG signal to 1000Hz by averaging the voltage values in sets of 30, and highpass filtered the down-sampled signal above 300Hz (*Travers and Norgren, 1986*; *Li et al., 2016*) using a $2^{nd}$ order Butterworth filter. The absolute value/magnitude of the filtered EMG signal was then lowpass filtered (again using a Butterworth filter of order 2) below 15Hz,

effectively capturing the envelope of variation of the EMG signal (plotted as the black curve in *Figure 4A1–A2*). This cutoff of 15Hz is sufficient for identifying orofacial behaviors, all of which occur at frequencies smaller than 10Hz (*Grill and Norgren, 1978a*; *Li et al., 2016*).

We subjected the envelope of the EMG signal to Bayesian spectrum analysis (BSA). BSA involves the construction of a probabilistic model of the generation of periodic signals from the superposition of sinusoids of different frequencies. We divided the signal on each trial into bins of width 300ms, with a step size of 1ms. We assumed that the EMG signal in each bin is produced by a sinusoid of a single frequency (plus noise) - in a probabilistic setting, this assumption implies the same model as a discrete-time Fourier transform. Contrary to the Fourier transform, however, BSA infers the posterior distribution of frequencies given the data. BSA has been shown to provide posterior estimates of frequencies that are an order of magnitude more precise than the Fourier transform (*Larry Bretthorst, 2013*; *Granqvist et al., 2011*). We used the BaSAR R package for BSA (*Granqvist et al., 2012*) and calculated the posterior probabilities of frequencies from 1Hz to 10Hz in 20 steps for each 300ms wide bin of data.

## Identifying the mean onset of aversive orofacial behavior

Rats respond to intra-oral deliveries of Qui in the concentration range used in our experiments ($10^{-4}$ to $10^{-3}$ M) with an initial set of non-specific investigative licks that are followed by large, jaw-opening mouth movements called gapes (*Grill and Norgren, 1978a*, *Figure 4A1–A2*). Gapes primarily involve activity of the anterior digastric muscle at 4–6Hz (*Grill and Norgren, 1978a*; *Li et al., 2016*) - we, therefore, used the probability of movements at 4–6Hz in the digastric EMG signal (from BSA, see previous section) as the probability of gaping ($\mathrm{Pr}_{Gape}$). This spectral measure of $\mathrm{Pr}_{Gape}$ has a strong correspondence with a previously-defined and above-discussed quadratic classifier (that tags individual mouth movements as gapes; *Li et al., 2016*). On individual Qui trials (*Figure 4B1–B2*), $\mathrm{Pr}_{Gape}$ from BSA is high (close to 1.0) when the quadratic classifier tags mouth movements as gapes. In addition, the average probability of gaping ($\overline{\mathrm{Pr}}_{Gape}$) from BSA (*Figure 4C1–C2*) is very similar to an across-trial, peri-stimulus average of the gapes picked by the quadratic classifier. In contrast to the quadratic classifier, however, the BSA measure of $\mathrm{Pr}_{Gape}$ is based entirely on the spectral content of the EMG signal. It, therefore, does not require the construction of a sufficiently complex classifier function (with a large enough set of experimenter-tagged examples to train the classifier) to pick out individual gapes. This also ensures that BSA considers bouts of movements together while calculating $\mathrm{Pr}_{Gape}$, making it robust against isolated large amplitude movements early in the animal's orofacial response. These initial movements were often found to be large licks on video and limited the accuracy of the quadratic classifier in *Li et al. (2016)* to 75%.

The probability of the transition from the rats' initial investigative licks to gapes depends on the concentration of Qui delivered: $10^{-3}$ M (Conc Qui) elicits gapes on more than twice the number of trials as $10^{-4}$ M (Dil Qui) (*Grill and Norgren, 1978a*; *Li et al., 2016*). Comparison of $\mathrm{Pr}_{Gape}$ on Dil and Conc Qui trials, thus, provides a natural way to calculate the mean onset of gaping across all the Qui trials in an experimental condition (again, Suc trials add little to this analysis, as the probability of 4–6Hz activity drops to 0 within 100-200msec of taste delivery). We expect the distribution of $\mathrm{Pr}_{Gape}$ on Dil Qui trials to be similar to that on Conc Qui trials in the investigative licking phase. Once gaping starts, however, we expect a large difference in the distributions of $\mathrm{Pr}_{Gape}$ on Dil and Conc Qui trials. $\mathrm{Pr}_{Gape}$ on Dil Qui trials, therefore, acts like a baseline for $\mathrm{Pr}_{Gape}$ on Conc Qui trials: we conclude that gapes have started only when $\mathrm{Pr}_{Gape}$ of Conc Qui begins to differ significantly from this baseline.

We used Beta distributions to describe $\mathrm{Pr}_{Gape}$ on Dil and Conc Qui trials. The Beta distribution is commonly used to model the probability parameter ($p$) of a Bernoulli (1/0) process and is expressed in terms of its two concentration parameters: $\alpha$ = observed number of 1s and $\beta$ = observed number of 0s. Gaping being a Bernoulli process, the Beta distribution is an appropriate choice for modeling $\mathrm{Pr}_{Gape}$. We defined one such Beta distribution in each time bin for Dil and Conc Qui separately, parametrized by the number of trials where the animal was gaping ($\mathrm{Pr}_{Gape}>0.5$) or not ($\mathrm{Pr}_{Gape}<0.5$). The Kullback-Leibler (KL) divergence of these Beta distributions ($\mathrm{D}_{KL}(\mathrm{ConcQui}||\mathrm{DilQui})$) provides a natural way to quantify the difference between $\mathrm{Pr}_{Gape}$ on Dil and Conc Qui trials - in general, the KL divergence between two Beta distributions with concentration parameters ($\alpha_1, \beta_1$) and ($\alpha_2, \beta_2$) can be written as:

$$D_{KL} = \log\Gamma(\sum_{j=1}^{j=2}\alpha_j) - \sum_{j=1}^{j=2}\log\Gamma(\alpha_j) - \log\Gamma(\sum_{j=1}^{j=2}\beta_j) + \sum_{j=1}^{j=2}\log\Gamma(\beta_j) + \sum_{j=1}^{j=2}(\alpha_j - \beta_j)(\psi(\alpha_j) - \psi(\sum_{j=1}^{j=2}\alpha_j)) \quad (6)$$

where $\Gamma$ and $\psi$ are the gamma and digamma functions respectively.

$D_{KL}(\text{ConcQui}||\text{DilQui})$ shows a sharp jump ~1s post taste delivery (*Figure 4E*), consistent with the timing of the transition from investigative licks to gapes (*Grill and Norgren, 1978a*; *Travers and Norgren, 1986*; *Li et al., 2016*). Finally, we calculated the cumulative sum of $D_{KL}(\text{ConcQui}||\text{DilQui})$ across time: the jump corresponding to the mean onset of gaping is expressed as a change in slope of the cumulative sum.

We could now fit two straight lines to the cumulative sum to capture this change in slope: the intersection of the two lines defines the mean timing of the onset of gaping (*Figure 4F*). We chose to instantiate this piecewise-linear regression as a Bayesian change-point analysis, where the change point captures the abrupt change in the slope of the cumulative sum of $D_{KL}(\text{ConcQui}||\text{DilQui})$. With a uniform prior on the timing of the change-point (from 0 to 2.0s post taste delivery), our model is as follows:

$$
\begin{aligned}
&\text{Intercept of first line} &&: \alpha_1 \sim \mathcal{N}(0,3) \\
&\text{Slope of first line} &&: \beta_1 \sim \mathcal{N}(0,1) \\
&\text{Intercept of second line} &&: \alpha_2 \sim \mathcal{N}(0,3) \\
&\text{Slope of second line} &&: \beta_2 \sim \mathcal{N}(0,1) \\
&\text{Prior on change point} &&: C \sim Uniform(0, 2000ms) \\
&\text{Prior on observation noise} &&: \sigma \sim Half\ Cauchy(0.5) \\
&\text{Mean cumsum of } D_{KL} &&: \overline{D_{KL}}(t) \sim \begin{cases} \alpha_1 + \beta_1 \times t, & t \leq C \\ \alpha_2 + \beta_2 \times t, & t > C \end{cases} \\
&\text{Cumsum of } D_{KL} &&: D_{KL}(\text{Conc Qui}||\text{Dil Qui})(t) \sim \mathcal{N}(\overline{D_{kl}}(t), \sigma)
\end{aligned}
\quad (7)
$$

The change-point, $C$, in this model is the mean onset of gaping. We inferred the posterior distribution of $C$ using MCMC sampling and performed significance tests using its 95% credible interval.

## Acknowledgements

This project was supported by the National Institutes of Health (NIH) R01 grants DC006666-00 and DC007703-06 (Donald Katz) and an International Predoctoral Fellowship from the Howard Hughes Medical Institute (HHMI) (Narendra Mukherjee). Computing resources for this project were provided by the National Science Foundation (NSF) XSEDE grant IBN180002 (Donald Katz) and by the high-performance computing cluster (HPCC) at Brandeis University. We would like to thank Dr. Paul Miller for invaluable feedback about this study and manuscript, and Shrabastee Banerjee and Dr. Jian-You Lin for many hours of consultation on building accurate and interpretive statistical models for our data.

## Additional information

### Funding

| Funder | Grant reference number | Author |
| --- | --- | --- |
| NIH Office of the Director | R01 DC006666-00 | Donald B Katz |
| National Science Foundation | IBN170002 | Donald B Katz |
| Howard Hughes Medical Institute | International Student Research Fellowship | Narendra Mukherjee |
| NIH Office of the Director | R01 DC007703-06 | Donald B Katz |
| National Science Foundation | IBN180002 | Donald B Katz |

The funders had no role in study design, data collection and interpretation, or the decision to submit the work for publication.

## Author contributions

Narendra Mukherjee, Conceptualization, Data curation, Software, Formal analysis, Funding acquisition, Validation, Investigation, Visualization, Methodology, Writing—original draft, Project administration, Writing—review and editing; Joseph Wachutka, Investigation, Methodology; Donald B Katz, Conceptualization, Resources, Formal analysis, Supervision, Funding acquisition, Validation, Investigation, Visualization, Methodology, Writing—original draft, Project administration, Writing—review and editing

## Author ORCIDs

Narendra Mukherjee (iD) https://orcid.org/0000-0003-3808-2622
Donald B Katz (iD) https://orcid.org/0000-0002-8444-6063

## Ethics

Animal experimentation: This study was performed in strict accordance with National Institute of Health guidelines and methods were approved in advance by the Brandeis University Institutional Animal Care and Use Committee in protocol numbers 15011 and 19002.

## Decision letter and Author response

Decision letter https://doi.org/10.7554/eLife.45968.011
Author response https://doi.org/10.7554/eLife.45968.012

## Additional files

### Supplementary files

• Transparent reporting form
DOI: https://doi.org/10.7554/eLife.45968.009

### Data availability

We have structured our electrophysiology datasets in a hierarchical data format (HDF5) and are hosting the files on a university-wide network share managed by Library and Technology Services (LTS) at Brandeis University. These HDF5 files contain our electrophysiology recordings, sorted spikes, single-neuron and population-level analyses (and associated plots and results). These files are prohibitively large to be hosted on a general-purpose fileshare platform - we request anyone interested in our datasets to contact the corresponding author, Donald Katz (dbkatz@brandeis.edu) who can put them in touch with LTS in order to create a guest account at Brandeis through which they can securely access our datasets (hosted on the katz-lab share at files.brandeis.edu).

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
