## [Decision Letter]

Thank you for submitting your article "Dynamical structure of cortical taste responses revealed by precisely-timed optogenetic perturbation" for consideration by *eLife*. Your article has been reviewed by four peer reviewers, including Arianna Maffei as the Reviewing Editor and Reviewer #1, and the evaluation has been overseen by Laura Colgin as the Senior Editor. The following individuals involved in review of your submission have agreed to reveal their identity: Asif A Ghazanfar (Reviewer #2); Justus Verhagen (Reviewer #3); Luca Mazzucato (Reviewer #4).

The reviewers have discussed the reviews with one another and the Reviewing Editor has drafted this decision to help you prepare a revised submission.

Summary:

There is general consensus that the manuscript by Mukherjee et al. provides a significant contribution to our understanding of how sensory perception drives action. Overall, the reviewers agreed that the experimental methods and the overall framework of the study are strong and that the results support the authors' conclusions. There are a few concerns regarding the description of the data and controls as well as a few suggestions to strengthen the paper's conclusions further by additional analyses.

Essential revisions:

A more descriptive title would be beneficial. The current title is pretty generic.

The description and justification of control conditions should be expanded. In a number of places in the manuscript there is lack of clarity as to which control is being referred: no opto, short opto, long opto. Moreover in Figure 5 there are a 2.5s control and a 0.5 control. A better way to refer to controls across the manuscript would be helpful, as would be a better description and justification for the choices of controls. An additional justification for why the authors chose to use light off control conditions instead of reporter-only viral injections should be provided.

Figure 3A would be more compelling if the regression coefficient analysis was reported also for 0.5s inactivation.

Figure 4 needs clarification as it is not clear what panel B1 and B2 are reporting. In addition, a description of Figure 6 is lacking in the Results.

Regarding the statistical methods used for the study, the authors spend a long portion of the methods describing and discussing Bayesian methods, but their analysis does not fully apply them to determine change points intervals. This would certainly strengthen the quantification of the transitions between epochs and provide a stronger quantification of the data. For example, one can assign a prior distribution on the intervals where change points C_I_ and C_P_ are concentrated according to the authors' definition in the first paragraph of the subsection “Modeling and change-point identification in ensemble firing data” (and Figure 6). These change points, if estimated for each session, would change the M-step, leaving the hard E-step unchanged. For clarity, it would also be useful to write down the full E-M equations and what modification of standard EM procedures lead to the "hard assignment" case used here.

The authors should perform the same CP analysis in Figure 7 to the other 2 conditions where laser ON occurred during 0-0.5s and 1.4-1.9s intervals. This is a necessary control to support the claim that only perturbations occurring at 0.7-1.2s affect gaping. Finally, in Figure 4 change point analysis would allow to better estimate when the cumulative sum of the KL divergence changes abruptly, under the hypothesis that the cumsum is piecewise linear.

---

## [Author Response]

Essential revisions:A more descriptive title would be beneficial. The current title is pretty generic.

We have changed the title of the paper to: “The impact of precisely-timed inhibition of gustatory cortex on taste behavior depends on single-trial ensemble response dynamics”

The description and justification of control conditions should be expanded. In a number of places in the manuscript there is lack of clarity as to which control is being referred: no opto, short opto, long opto. Moreover in Figure 5 there are a 2.5s control and a 0.5 control. A better way to refer to controls across the manuscript would be helpful, as would be a better description and justification for the choices of controls. An additional justification for why the authors chose to use light off control conditions instead of reporter-only viral injections should be provided.

The reviewers and editors bring up a set of very good inter-related points. We have substantially cleaned up and clarified the manuscript with regard to this issue. Included in this cleanup is an explicit statement (subsection “Experimental paradigm and data overview”, second paragraphand subsection **“**Early 0.5s perturbations delay single-neuron palatability-related responses while late 0.5s perturbations do not”, last paragraph) that all comparisons were within-session (i.e., eachlaser condition is only directly compared to other laser conditions and no-laser trials collected in the same session); thus we are able to more simply handle the controls, simply as “no-laser trials.” We have also significantly streamlined and reduced the space given to the 2.5s laser sessions (note changes to Figure 5), which are almost entirely (with noted exceptions) peripheral to the main results of our manuscript.

Finally, we set up and maintain throughout the document a simpler, more concrete nomenclature (“0.5s perturbations” vs. “2.5s perturbations;”), and explain (subsection “Experimental paradigm and data overview”, second paragraph; subsection “Early 0.5s perturbations delay single-neuron palatability-related responses while late 0.5s perturbations do not” and subsection “Data analysis”)why in the case of our current work a reporter-only virus control is of little or no value – specifically because our conclusions rely on comparisons of different laser timings within the same rat and session: had the archT virus had some confounding impact, that impact would have affected each trial type equivalently, such that we would have seen no within-session differences; given the fact that we saw strong within-session differences, our conclusion that the impact of GC perturbation depends on precisely when the perturbation occurs is valid regardless.

Figure 3A would be more compelling if the regression coefficient analysis was reported also for 0.5s inactivation.

We have, as requested, expanded Figure 3 – adding new panels that illustrate the regression coefficient analysis brought to bear on the 0.5s inactivation trials (Figure 3A1-3). The results of these analyses, while noisy (see below), are entirely in accord with our hypotheses.

We hasten to point out (and have done so in the caption of Figure 3 and in the text in subsections “GC perturbation impacts orofacial behavior only if delivered before the coherent ensemble transition to palatability-related activity”, last paragraph and **“**Characterizing the time course of the palatability index”), however, that these analyses should be interpreted with great caution. One basic assumption of curve fitting (here, the sigmoid) is stationarity – the expectation that the data (in this case, spiking) were sampled from conditions that remained stable, and that these data can therefore be reasonably thought to follow a single function. The 0.5s perturbations shred this assumption: the condition of the neural network changes suddenly with the onset of inhibition, and then again with the offset of inhibition, such that it is unreasonable to expect a single regression to describe firing across laser onset and offset; it is similarly unreasonable to expect the function after offset to be a simple continuation of the function prior to onset.

We considered performing piecemeal fitting (separately analyzing pre-, peri-, and post-laser data), but the short (0.5s) duration of the perturbations makes this analysis prohibitively low-powered – particularly given the fact, made abundantly clear in this and previous studies, that the stimulus-induced evolution of neural firing differs in timing from trial to trial.

We chose to fit single sigmoids to the palatability regression coefficients over time as the “lesser of two evils,” and then to be explicitly circumspect in our interpretation of these analyses. As part of this change, we make several explicit points:

1) We note that this is a valid analysis for control and 2.5s perturbation trials, as both involve 2.5s of firing in a single optogenetic condition (laser, no laser).

2) We note that Figure 3C and D present the most robust features of this analysis – features that support our ultimate conclusions.

3) Finally, we note in the Discussion that our demonstration of a strong causal role of single trial ensemble dynamics in consumption behavior supersedes the trial-averaging analyses used in Figure 3.

Figure 4 needs clarification as it is not clear what panel B1 and B2 are reporting.

Additional clarification has been added to the text (subsection “Measuring aversive orofacial behaviors (gapes)”) and to the Figure 4 legend.

In addition, a description of Figure 6 is lacking in the Results.

We apologize for that oversight, which has now been corrected (subsection “Modeling and change-point identification in ensemble firing data”).

Regarding the statistical methods used for the study, the authors spend a long portion of the methods describing and discussing Bayesian methods, but their analysis does not fully apply them to determine change points intervals. This would certainly strengthen the quantification of the transitions between epochs and provide a stronger quantification of the data. For example, one can assign a prior distribution on the intervals where change points C_I_ and C_P_ are concentrated according to the authors' definition in the first paragraph of the subsection “Modeling and change-point identification in ensemble firing data” (and Figure 6). These change points, if estimated for each session, would change the M-step, leaving the hard E-step unchanged.

We apologize for not more fully clarifying our use of Bayesian methods in the change-point analysis. In fact, the change-point EM algorithm assigns prior distributions on the intervals for C_I_ and C_P_ exactly as the reviewer suggests – specifically, we assigned uniform priors on the intervals defining C_I_ and C_P_. The E step of the EM algorithm then becomes a fairly simple calculation of the data likelihood for each possible pair of change points, given the emission parameters. The uniform prior assigns equal probability of each C_I_, C_P_ pair a priori, thusallowing us to pick the pair that maximizes the data likelihood to complete the “hard assignment” E step of our algorithm.

The reviewer is correct in noting a difference, however: in contrast to the other Bayesian analyses in the paper, we use EM rather than MCMC sampling in the change-point analysis. We do so for a straight-forward reason: the specification of 2 change points in the model results in a posterior that has 2 modes, corresponding to different (but equivalent) “labels” for the change points (one mode labels the first change point as CP1 and the second as CP2, and the other switches the ordering of the change points); the 2 modes are equivalent solutions to the change-point model, and picking any one of them is a valid solution. MCMC sampling, however, jumps between these 2 solutions and fails to yield a convergent chain – in the literature, this is known as the “label switching” problem. As (properly initialized) EM converges to one of the 2 equivalent solutions, avoiding this label switching problem, we use it for the change-point analysis, which is the only analysis in which this problem arises in our paper. This logic is now clarified in the Materials and methods section.

For clarity, it would also be useful to write down the full E-M equations and what modification of standard EM procedures lead to the "hard assignment" case used here.

In the revised manuscript, we have, as requested, expanded and rewritten the full E-M equations for our EM algorithm (subsection “Modeling and change-point identification in ensemble firing data”).

The authors should perform the same CP analysis in Figure 7 to the other 2 conditions where laser ON occurred during 0-0.5s and 1.4-1.9s intervals. This is a necessary control to support the claim that only perturbations occurring at 0.7-1.2s affect gaping.

It is important that we note that the central claim of the paper is *not* that perturbations in a particular fixed window of time have a particular impact; rather, it is that perturbations beginning beforean internally-determined ensemble firing-rate transition (the palatability change point) delay gaping. The latency of this transition varies from trial to trial, and thus while the “middle 0.5s perturbation” frequently delays gaping, it often does not (see Figure 7B and C), and while the “late 0.5s perturbation” usually has no impact on gaping, once in a while it should (when the GC ensemble transition occurs particularly late, see below). Our precise thesis is now stated more clearly and extensively in subsection “GC perturbation delays the onset of aversive orofacial behavior”.

The early 0.5s perturbations necessarily arrive before the palatability change point occurs, and, consistent with our thesis, necessarily delay the onset of gaping (see Figure 5A). Alas, activity following the perturbation is inevitably difficult to interpret, and difficult to compare to control trials containing unperturbed activity.

We absolutely see the value in the reviewers’ suggestion that we perform CP analysis on all pre-laser activity in the study, however. For the revision, we now provide those analyses of the late 0.5s perturbation trials; these analyses further support our thesis, in that when the 12% of these trials in which the GC ensemble transition had not yet occurred are removed, the small but significant delay in gaping in these trials (Figure 5) totally vanishes. These results are now described in subsection “GC perturbation impacts orofacial behavior only if delivered before the coherent ensemble transition to palatability-related activity”.

Finally, in Figure 4 change point analysis would allow to better estimate when the cumulative sum of the KL divergence changes abruptly, under the hypothesis that the cumsum is piecewise linear.

We apologize again for having left unclear our use of EM and MCMC in change-point analyses. In fact, and just as the reviewers advise, we did implement a change-point model to detect the abrupt change in the cumulative sum of the KL divergence of gaping probabilities; we more clearly describe it as such in the revision. We were then able to use MCMC for inference in this particular change-point model, because it consisted of a single change point (and thus we not run into the label switching problem described in our sixth response). We have clarified these details in the Materials and methods section.